# Bandits with Ranking Feedback

**Davide Maran**[*]
Politecnico di Milano
davide.maran@polimi.it

**Francesco Bacchiocchi**[*]
Politecnico di Milano
francesco.bacchiocchi@polimi.it

**Francesco Emanuele Stradi**[*]
Politecnico di Milano
francescoemanuele.stradi@polimi.it

**Matteo Castiglioni**
Politecnico di Milano
matteo.castiglioni@polimi.it

**Nicola Gatti**
Politecnico di Milano
nicola.gatti@polimi.it

**Marcello Restelli**
Politecnico di Milano
marcello.restelli@polimi.it

## Abstract

In this paper, we introduce a novel variation of multi-armed bandits called *bandits with ranking feedback*. Unlike traditional bandits, this variation provides feedback to the learner that allows them to rank the arms based on previous pulls, without quantifying numerically the difference in performance. This type of feedback is well-suited for scenarios where the arms' values cannot be precisely measured using metrics such as monetary scores, probabilities, or occurrences. Common examples include human preferences in matchmaking problems. Furthermore, its investigation answers the theoretical question on how numerical rewards are crucial in bandit settings. In particular, we study the problem of designing *no-regret* algorithms with ranking feedback both in the *stochastic* and *adversarial* settings. We show that, with stochastic rewards, differently from what happens with non-ranking feedback, no algorithm can suffer a logarithmic regret in the time horizon $T$ in the instance-dependent case. Furthermore, we provide two algorithms. The first, namely DREE, guarantees a superlogarithmic regret in $T$ in the instance-dependent case thus matching our lower bound, while the second, namely R-LPE, guarantees a regret of $\widetilde{\mathcal{O}}(\sqrt{T})$ in the instance-independent case. Remarkably, we show that no algorithm can have an optimal regret bound in both instance-dependent and instance-independent cases. Finally, we prove that no algorithm can achieve a sublinear regret when the rewards are adversarial.

## 1 Introduction

*Multi-armed bandits* are well-known sequential decision-making problems where a learner is given a number of arms whose reward is unknown [Lattimore and Szepesvari, 2017]. At every round, the learner can pull an arm and observe a realization of the reward associated with that arm, which can be generated *stochastically* [Auer et al., 2002] or *adversarially* [Auer et al., 1995]. The central question

---

[*]Equal Contribution.

38th Conference on Neural Information Processing Systems (NeurIPS 2024).

in multi-armed bandits concerns how to address the *exploration/exploitation* tradeoff to minimize the *regret* between the reward provided by the *learning policy* and the optimal *clairvoyant* algorithm.

In this paper, we introduce a novel variation of multi-armed bandits that, to the best of our knowledge, is unexplored so far. We name the model as *bandits with ranking feedback*. This feedback provides the learner with a partial observation over the rewards given by the arms. More precisely, the learner can rank the arms based on the previous pulls they experienced, but they cannot quantify numerically the difference in performance. Thus, the learner is not allowed to asses how much an arm is better or worse than another. This type of feedback is well-suited for scenarios where the arms' values cannot be precisely measured using metrics such as monetary scores, probabilities, or occurrences, and naturally applies to various settings, *e.g.*, when dealing with human preferences such as in matchmaking settings among humans and when the scores cannot be revealed for privacy or security reasons. This latter case can be found, *e.g.*, in online advertising platforms offering automatic bidding services as they have no information on the actual revenue of the advertising campaigns since the advertisers prefer not to reveal these values being sensible data for the companies. Notice that a platform can observe the number of clicks received by an advertising campaign, but it cannot observe the revenue associated with that campaign. Remarkably, our model poses the interesting theoretical question whether the lack of numerical scores precludes the design of sublinear regret algorithms or worsens the regret bounds that are achievable when numerical scores are available.

**Original contributions.** We study the problem of designing *no-regret* algorithms for the bandits with ranking feedback problem, both in *stochastic* and *adversarial* settings. In the case of adversarial rewards, we prove that no algorithm can achieve sublinear regret. In contrast, with stochastic rewards, we show that the ranking feedback does not preclude such a possibility. In particular, in the instance-dependent case, we show that no algorithm can achieve logarithmic regret in the time horizon, and we provide an algorithm, namely DREE (Dynamical Ranking Exploration-Exploitation), guaranteeing a regret bound that matches this lower bound. In the instance-independent case, a crucial question is whether there exists an algorithm providing a better regret bound compared to the one achieved by the well-known Explore-then-Commit algorithm, which trivially guarantees an $\mathcal{O}(T^{2/3})$ regret upper bound. We positively answer this question by designing an algorithm, namely R-LPE (Ranking Logarithmic Phased Elimination), which guarantees a regret of $\widetilde{\mathcal{O}}(\sqrt{T})$ in the instance-independent case if the rewards are Gaussian. To achieve this result, we derive several non-standard results that allow us to discretize Brownian motions, which are of independent interest. These two different approaches leave open the problem of whether there exists an algorithm achieving optimal performance in both the instance-dependent and instance-independent cases. We negatively answer this question by showing that no algorithm can achieve an optimal regret bound in both cases, confirming the need to design two distinct algorithms for the two cases. Finally, we numerically evaluate our DREE and R-LPE algorithms in a testbed, and we compare their performance with some baselines from the literature in different settings. We show that our algorithms dramatically outperform the baselines in terms of empirical regret.

**Related works.** The field most related to bandits with ranking is *preference learning*, which aims at learning the preferences of one or more agents from some observations [Fürnkranz and Hüllermeier, 2010]. Let us remark that preference learning has recently gained a lot of attention from the scientific community, as it enables the design of AI artifacts capable of interacting with human-in-the-loop (HTL) environments. Indeed, human feedback may be quite misleading when it is asked to report numerical values, while humans are far more effective at reporting ranking preferences. The preference learning literature mainly focuses on two kinds of preference observations: pairwise preferences and ranking. In the first case, the data observed by the learner involves preferences between two objects, *i.e.*, a partial preference is given to the learner. In the latter, a complete ranking of the available data is given as feedback. Our work belongs to the latter branch. Preference learning has been widely investigated by the online learning community, see, *e.g.*, [Bengs et al., 2018].

Precisely, our work presents several similarities with the *dueling bandits* settings [Yue et al., 2012, Saha and Gaillard, 2022, Lekang and Lamperski, 2019], where, in each round, the learner pulls two arms and observes a ranking over them. Nevertheless, although dueling bandits share similarities to our setting, they present substantial differences. Specifically, in our model, the learner observes a ranking depending on the arms they have pulled so far. In dueling bandits, the learner observes an instantaneous comparison between the arms they have just pulled; thus, the outcome of such a comparison does not depend on the arms previously selected, as is the case of bandits with ranking

feedback. As a consequence, while in bandits with ranking feedback the goal of the learner is to exploit the arm with the highest mean, in dueling bandits the goal of the learner is to select the arm winning with the highest probability. Furthermore, while we adopt the classical notion of regret used in the bandit literature to assess the theoretical properties of our algorithms, in dueling bandits, the algorithms are often evaluated with a suitable notion of regret, which differs from the classical one.

Dueling bandits have their reinforcement learning (RL) counterpart in the *preference-based reinforcement learning* (PbRL), see, *e.g.*, [Novoseller et al., 2019] and [Wirth et al., 2017]. Interestingly, PbRL techniques differ from the standard RL approaches in that they allow an algorithm to learn from non-numerical rewards; this is particularly useful when the environment encompasses human-like entities [Chen et al., 2022]. Furthermore, *preference-based reinforcement learning* provides a bundle of results, ranging from theory [Xu et al., 2020] to practice [Christiano et al., 2017, Lee et al., 2021]. In PbRL, preferences may concern both states and actions; contrariwise, our framework is stateless since the rewards gained depend only on the action taken during the learning dynamic. Moreover, the differences outlined between dueling bandits and bandits with ranking feedback still hold for preference-based reinforcement learning, as preferences are considered between observations instead of the empirical mean of the accumulated rewards.

**Paper structure.**    The paper is structured as follows. In Section 2, we report the problem formulation, the setting, and the necessary notation. In Section 3, we study the stochastic setting, that is, when the rewards are sampled from a fixed distribution. In Section 3.1, we present an instance-dependent regret lower bound that characterizes our problem. Thus, in Section 3.2, we propose our algorithm and we show that it achieves a tight instance-dependent regret bound. In Section 3.3, we show a trade-off between the regret upper bound achievable by any algorithm in the instance-dependent and instance-independent cases. Finally, in Section 3.4, we present an algorithm achieving an optimal instance-independent regret bound when the rewards are Gaussian. In Section 4, we study the adversarial setting, that is, when no statistical assumptions are made about the rewards. In this case, we present our impossibility result, showing that no algorithm can achieve sublinear regret. In Appendix A and Appendix B, we report the omitted proofs for the stochastic setting. In Appendix C, we report the omitted proof for the adversarial setting. Finally, in Appendix D, we report the empirical evaluation of our algorithms.

## 2   Problem formulation

In this section, we formally state the model of bandits with ranking feedback and discuss the learner-environment interaction. Subsequently, we define policies and the notion of regret both in the *stochastic* and in the *adversarial* setting.

**Setting and interaction.**    Differently from the classical version of the multi-armed bandit problem—see, *e.g.*, [Lattimore and Szepesvari, 2017]—in which the learner observes the *reward* associated with the pulled arm, in bandits with ranking feedback the learner can only observe a *ranking* over the arms based on the previous pulls. Formally, we assume the learner-environment interaction to unfold as follows.[2]

   (i) At every round $t \in [T]$, where $T$ is the time horizon, the learner chooses an arm $i_t \in \mathcal{A} :=$ $[n]$, where $\mathcal{A}$ is the set of available arms and $n = |\mathcal{A}| < +\infty$.

  (ii) We study both stochastic and adversarial settings. In the stochastic setting, the environment draws the reward $r_t(i_t)$ associated with arm $i_t$ from a probability distribution $\nu_{i_t}$, *i.e.*, $r_t(i_t) \sim \nu_{i_t}$, whereas, in the adversarial setting, $r_t(i_t)$ is chosen adversarially by an opponent from a bounded set of reward functions.

 (iii) There is a bandit feedback on the reward of the arm $i_t \in \mathcal{A}$ pulled at round $t$ leading to the estimate of the empirical mean of $i_t$ as follows:

$$\hat{r}_t(i) := \frac{\sum_{j \in \mathcal{W}_t(i)} r_j(i)}{Z_i(t)},$$

---

[2]Given $n \in \mathbb{N}^*$ we denote with $[n] := \{1, \ldots, n\}$. Furthermore, given a finite set $\mathcal{A}$, we denote by $\mathcal{S}_\mathcal{A}$ the set containing all the possible permutations of its elements. Similarly, we let $\mathcal{P}(\mathcal{A})$ be the set containing all the probability distributions with support $\mathcal{A}$.

where $\mathcal{W}_t(i) := \{\tau \in [t] \mid i_\tau = i\}$ and $Z_i(t) := |\mathcal{W}_t(i)|$.[3] The learner observes the ranking over the empirical means $\{\hat{r}_t(i)\}_{i\in\mathcal{A}}$. Formally, we assume that the ranking $\mathcal{R}_t \in \mathcal{S}_\mathcal{A}$ observed by the learner at round $t$ is such that:

$$\hat{r}_t(\mathcal{R}_{t,i}) \geq \hat{r}_t(\mathcal{R}_{t,j}) \ \forall t \in [T] \ \forall i,j \in [n] \text{ s.t. } i \geq j,$$

where $\mathcal{R}_{t,i} \in \mathcal{A}$ denotes the $i$-th element in the ranking $\mathcal{R}_t$ at round $t \in [T]$. Ties are broken in favor of the lower index.

For the sake of clarity, we provide an example to illustrate our setting and the corresponding learner-environment interaction.

**Example.** We consider an instance with two arms, *i.e.*, $\mathcal{A} = \{1,2\}$, where the learner plays the first arm at rounds $t = 1$ and $t = 3$, and the second arm at round $t = 2$, such that $\mathcal{W}_3(1) = \{1,3\}$ and $\mathcal{W}_3(2) = \{2\}$. Let $r_1(1) = 1$ and $r_3(1) = 5$ be the rewards obtained from playing the first arm at rounds $t = 1$ and $t = 3$, respectively, and let $r_2(2) = 5$ be the reward for playing the second arm at round $t = 2$. The empirical means of the two arms and the resulting rankings at each round $t \in [3]$ are given by:

$$\begin{cases} \hat{r}_t(1) = 1, \ \hat{r}_t(2) = 0 & \mathcal{R}_t = \langle 1,2 \rangle & t = 1 \\ \hat{r}_t(1) = 1, \ \hat{r}_t(2) = 5 & \mathcal{R}_t = \langle 2,1 \rangle & t = 2 \\ \hat{r}_t(1) = 3, \ \hat{r}_t(2) = 5 & \mathcal{R}_t = \langle 2,1 \rangle & t = 3. \end{cases}$$

**Policies and regret.** At every round $t$, the arm played by the learner is prescribed by a policy $\pi$. In both the stochastic and adversarial settings, we let the policy $\pi$ be a randomized map from the history of the interaction $H_{t-1} = (\mathcal{R}_1, i_1, \mathcal{R}_2, i_2, \ldots, \mathcal{R}_{t-1}, i_{t-1})$ to the set of all probability distributions with support $\mathcal{A}$. Formally, we let $\pi : H_{t-1} \to \mathcal{P}(\mathcal{A})$, for $t \in [T]$, such that $i_t \sim \pi(H_{t-1})$. As is customary in stochastic bandits, the learner's goal is to design a policy $\pi$ that minimizes the cumulative expected regret, which is formally defined as follows:

$$R_T(\pi) = \mathbb{E}\left[\sum_{t=1}^{T} r_t(i^*) - r_t(i_t)\right],$$

where the expectation in the definition of regret is taken over the randomness of both the policy and the environment, and we define $i^* \in \arg\max_{i\in\mathcal{A}} \mu_i$ with $\mu_i := \mathbb{E}[\nu_i]$ for each $i \in [n]$. In contrast, in the adversarial setting, the regret is defined as $R_T(\pi) = \sum_{t=1}^{T} r_t(i^*) - r_t(i_t)$ and we define $i^* \in \arg\max_{i\in\mathcal{A}} \sum_{t=1}^{T} r_t(i)$. Furthermore, from here on, we omit the dependence on the policy selected by the learner $\pi$ in the regret formulation, referring to $R_T(\pi)$ as $R_T$ whenever it is clear from the context. In the stochastic setting, we introduce the following additional notation. Given an arm $i \in [n]$, we let $\Delta_i := \mu_{i^*} - \mu_i$ represent the suboptimality gap of that arm. Furthermore, when $n = 2$, we simply refer to the suboptimality gap as $\Delta$.

As we will further discuss, the impossibility of observing the reward realizations raises several technical challenges when designing no-regret algorithms, as the approaches adopted for standard (non-ranking) bandits may be challenging to apply within our framework. In the following sections, we discuss how the lack of this information degrades the performance of the algorithms when the feedback is provided as a ranking.

## 3 Analysis in the stochastic setting

As a preliminary observation, we note that optimistic approaches, such as the UCB1 algorithm, are challenging to apply within our framework. This is because the learner lacks the information to estimate the expected rewards of the different arms, making it difficult to infer confidence bounds. Therefore, the most popular algorithm one can employ in bandits with ranking feedback is the *explore-then-commit* (EC) algorithm (see, *e.g.*, Auer et al. [2002]), in which the learner explores the arms uniformly during the initial rounds and then commits to the one with the highest empirical mean. However, as we will also discuss in the following, such algorithm achieve suboptimal regret guarantees. Thus, in the rest of this section, we present two algorithms specifically designed to achieve optimal instance-dependent and instance-independent regret bounds.

---

[3]Note that the latter definition is well-posed as long as $|\mathcal{W}_t(i)| > 0$. For each $i \in \mathcal{A}$ and $t \in [T]$ such that $|\mathcal{W}_t(i)| = 0$, we let $\hat{r}_t(i) = 0$.

## 3.1 Instance-dependent lower bound

In the classical multi-armed bandit problem, it is well known that it is possible to achieve an instance-dependent regret bound that is logarithmic in the time horizon $T$. However, in this section, we show that such a result does not hold when the feedback is provided as a ranking. Our impossibility result leverages a connection between random walks and the cumulative rewards of the arms. Formally, we define an (asymmetric) random walk as follows.

**Definition 1.** *A random walk is a stochastic process $\{G_t\}_{t \in \mathbb{N}}$ such that:*

$$G_t = \begin{cases} 0 & t = 0 \\ G_{t-1} + \epsilon_t & t \geq 1 \end{cases},$$

*where $\{\epsilon_t\}_{t \in \mathbb{N}}$ is an i.i.d. sequence of integrable random variables, and $\mathbb{E}[\epsilon_t] = p$ is the drift of the random walk.*

Before introducing our negative result, we introduce a lemma that characterizes the average performance of two random walks with different drifts.

**Lemma 1** (Separation lemma). *Let $\{G_t\}_{t \in \mathbb{N}}, \{G'_t\}_{t \in \mathbb{N}}$ be two independent random walks defined as:*

$$G_{t+1} = G_t + \epsilon_t \qquad and \qquad G'_{t+1} = G'_t + \epsilon'_t,$$

*where $G_0 = G'_0 = 0$ and the drifts satisfy $\mathbb{E}[\epsilon_t] = p > q = \mathbb{E}[\epsilon'_t]$, for each $t \in \mathbb{N}$. Then, we have:*

$$\mathbb{P}\Big(\forall t, t' \in \mathbb{N}^* \ \ G_t/t \geq G'_{t'}/t'\Big) \geq c(p, q) > 0.$$

We remark that the exact value of the constant $c(p, q)$ in Lemma 1 depends only on the two drifts $p$ and $q$, as well as the probability distribution defining these drifts. In the simpler case of Bernoulli distributions, the constant $c(p, q)$ can be derived in closed form, as shown in Lemma 3 in the appendix. The rationale behind Lemma 1 is that, when considering two random walks with different drifts, there exists a *separating line* between them with strictly positive probability. Thus, with non-negligible probability, the empirical mean of the process with the higher drift always upper bounds the empirical mean of the process with the lower drift.

We also observe that the cumulative reward collected by a specific arm during the learning process can be represented as a random walk, whose drift corresponds to the expected reward associated with that arm. In the classical version of the multi-armed bandit problem, the learner can fully observe the evolution of this random walk, while in our setting, the learner can only observe a ranking of the different arms, making it impossible to quantify their performance numerically. Therefore, in bandits with ranking feedback, the only way for the learner to assess how close two arms are in terms of expected reward is by observing subsequent switches in their positions within the ranking. However, Lemma 1 implies that even if two arms have very close expected rewards, the learner may never observe a switch in the ranking, as the average mean of the arm with the higher expected reward may upper bound the second throughout the entire learning process. As a result, the following negative result holds.

**Theorem 1** (Instance-dependent lower bound). *Let $\pi$ be any policy for the bandits with ranking feedback problem, then, for any $C : [0, +\infty) \to [0, +\infty)$, there exists a $\Delta > 0$ and a time horizon $T > 0$ such that $R_T > C(\Delta) \log(T)$.*

## 3.2 Instance-dependent upper bound

We introduce the Dynamical Ranking Exploration-Exploitation algorithm (DREE). The pseudo-code is provided in Algorithm 1. As usual in bandit algorithms, in the first $n$ rounds, a pull for each arm is performed (Lines 2–3). At every subsequent round $t > n$, the exploitation/exploration trade-off is addressed by playing the best arm according to the received feedback unless there is at least one arm whose number of pulls at $t$ is smaller than a superlogarithmic function $f(t) : (0, \infty) \to \mathbb{R}_+$.[4] More precisely, the algorithm plays an arm $i$ at round $t$ if it has been pulled less than $f(t)$ times

---

[4]A function $f(t)$ is superlogarithmic when $\liminf\limits_{t \to \infty} \frac{f(t)}{\log(t)} = +\infty$. Coherently, a subpolynomial function is such that for every $\alpha > 0$ we have $\limsup\limits_{t \to \infty} \frac{f(t)}{t^\alpha} = 0$.

(Lines 4–5), where ties due to multiple arms pulled less than $f(t)$ times are broken arbitrarily. Instead, if all arms have been pulled at least $f(t)$ times, the arm in the highest position of the last ranking feedback is pulled (Lines 6–8). Each round terminates with the learner receiving the updated ranking over the arms as feedback (Line 9). Let us observe that the exploration strategy of Algorithm 1 is deterministic, and the only source of randomness concerns the realization of the arms' rewards.

We state the following result, providing the regret upper bound of Algorithm 1 as a function of $f$.

**Theorem 2** (Instance-dependent upper bound). *Assume that the reward distribution of every arm is 1-subgaussian. Let $f : (0, \infty) \to \mathbb{R}_+$ be a superlogarithmic nondecreasing function in t. Then there is a term $C(f, \Delta_i)$ for each sub-optimal arm $i \in [n]$ which does not depend on $T$, such that Algorithm 1 satisfies $R_T \leq (1 + f(T)) \sum_{i=1}^{n} \Delta_i + \log(T) \sum_{i=1}^{n} C(f, \Delta_i)$.*

To minimize the asymptotic dependence on $T$ of the cumulative regret suffered by the algorithm, we can choose as an example $f(t) = \log(t)^{1+\delta}$, where the parameter $\delta > 0$ is as small as possible. However, if $\Delta_i < 1$, the minimization of $\delta$ comes at the cost of increasing the term $C(f, \Delta_i)$ as it grows exponentially as $\delta$ goes to zero. In particular, the terms $C(f, \Delta_i)$ are formally defined in the following corollary.

**Corollary 1.** *Let $\delta > 0$ and $f(t) = \log(t)^{1+\delta}$ be the sperlogarithmic function used in Algorithm 1, then we have:*

$$C(f, \Delta_i) = \frac{2\Delta_i \left( e^{\left( (2/\Delta_i^2)^{1/\delta} \right)} + 1 \right)}{1 - e^{-\Delta_i^2/2}}.$$

We remark that the term $C(f, \Delta_i)$ depends exponentially on $\Delta_i$, suggesting that $C(f, \Delta_i)$ may be large even when adopting values of $\delta$ that are not arbitrarily close to zero. At this point, a natural question arises. Is such an exponential dependence with respect to the suboptimality gaps unavoidable, or is it a consequence of the kind of feedback the learner receives? In the next section, we answer this question.

Furthermore, let us observe that Algorithm 1 satisfies important properties.

**Algorithm 1** Dynamical Ranking Exploration-Exploitation (DREE)

---
1: **for** $t \in [T]$ **do**
2:      **if** $t \leq n$ **then**
3:          play arm $i_t$
4:      **else if** arm $i$ has played $< f(t)$ times **then**
5:          play $i_t = i$
6:      **else**
7:          play $i_t = \mathcal{R}_{t-1,1}$
8:      **end if**
9:      receive updated ranking $\mathcal{R}_t$
10: **end for**

---

More precisely, (i) it matches the instance-dependent regret lower-bound, since $f(\cdot)$ can be chosen arbitrarily close to $\log(t)$, (ii) it works without requiring the knowledge of the time horizon $T$, thus being an *any-time algorithm*.

### 3.3 Instance dependent/independent trade-off

In this section, we provide a negative result, showing that *no algorithm* can achieve good performance in terms of both instance-dependent and instance-independent regret bounds, suggesting that the two cases need to be studied separately. Formally, the following theorem holds.

**Theorem 3** (Instance Dependent/Independent Trade-off). *Let $\pi$ be any policy for the bandits with ranking feedback problem. If $\pi$ satisfies the following properties:*

- *(instance-dependent regret upper bound) $R_T \leq \sum_{i=1}^{n} C(\Delta_i) T^\alpha$*

- *(instance-independent regret upper bound) $R_T \leq nCT^\beta$*

*then, $2\alpha + \beta \geq 1$, where $\alpha, \beta \geq 0$.*

*Proof.* (Sketch) Let $p_1 = 0.5$, $p_2 = 0.5 - \Delta$, $p_2^* = 0.5 + \Delta$, for some $\Delta > 0$ specified in the proof. We consider three instances:

$$\mathcal{P} : \begin{cases} \nu_1 = Be(p_1) \\ \nu_2 = Be(p_2) \end{cases} \qquad \mathcal{P}^* : \begin{cases} \nu_1 = Be(p_1) \\ \nu_2 = Be(p_2^*) \end{cases} \qquad \mathcal{P}^{**} : \begin{cases} \nu_1 = Be(1) \\ \nu_2 = Be(0) \end{cases}$$

Clearly, the first arm is optimal in instances $\mathcal{P}, \mathcal{P}^{**}$, while the second arm is optimal in $\mathcal{P}^*$. We then define the event:

$$E_t = \bigcap_{\tau=1}^{t} \{\mathcal{R}_t = \langle 1, 2 \rangle\}.$$

In the first and in the third instances, the event $E_t$ corresponds to the optimal arm being ranked in the first position for the first $t$ time steps, while the opposite holds for the second instance. We observe the following two key points. (1) Since the event $E_t$ contains the realization of all the rankings up to time $t$, no policy can distinguish between the three instances until this event holds. Therefore, we have:

$$\mathbb{E}_{\mathcal{P}}[Z_2(t)|E_t] = \mathbb{E}_{\mathcal{P}^*}[Z_2(t)|E_t] = \mathbb{E}_{\mathcal{P}^{**}}[Z_2(t)|E_t].$$

(2) In instance $\mathcal{P}^{**}$, the event $E_t$ happens almost surely for every $t$. Therefore, to ensure that the policy $\pi$ achieves an instance-dependent regret bounded by $C(1)T^\alpha$ in this instance, we need $\mathbb{E}[Z_2(T)|E_T] \leq CT^\alpha$ in all three instances.

The main question at this point reduces to: are $CT^\alpha$ pulls of the last-ranking arm enough to distinguish between the two instances $\mathcal{P}$ and $\mathcal{P}^*$? With a change of measure argument *restricted to the first $CT^\alpha$ pulls of the last-ranking arm*, we are able to show that, for a sufficiently small value of $\Delta > 0$, distinguishing between $\mathcal{P}, \mathcal{P}^*$ is impossible with strictly positive probability. Then, we can prove that if the previous consideration holds, in the instance $\mathcal{P}^*$, we have:

$$R_T \geq \Omega\left(T^{1-2\rho\alpha}\right),$$

for a constant $\rho > 0$ close to one. This lower bound on the instance-independent regret entails that $\beta \geq 1 - 2\alpha$, or, equivalently $\beta + 2\alpha \geq 1$. $\qquad\square$

From Theorem 3, we can easily infer the following impossibility result.

**Corollary 2.** *There is no policy $\pi$ for the bandits with ranking feedback problem achieving both subpolynomial regret in the instance-dependent case, i.e., for every $\alpha > 0$, there exists a function $C(\cdot)$ such that $R_T \leq \sum_{i=1}^{n} C(\Delta_i)T^\alpha$, and sublinear regret in the instance-independent case.*

To ease the interpretation of Corollary 2, in the following result, we discuss the instance-independent regret bound achieved by Algorithm 1.

**Corollary 3.** *For every choice of $\delta > 0$ in $f(t) = \log(t)^{1+\delta}$, there is no value of $\eta > 0$ for which Algorithm 1 achieves an instance-independent regret bound of the form $R_T \leq \mathcal{O}(T^{1-\eta})$.*

The above result shows that Algorithm 1 suffers from linear instance-independent regret in $T$, except for logarithmic terms. Moreover, the following corollary of Theorem 3 answers the question raised in the previous section. Indeed, we can prove that the unpleasant dependence on the suboptimality gaps $\Delta_i$ is not a feature of Algorithm 1; instead, it cannot be avoided until the instance-dependent regret has a good order in $T$.

**Corollary 4.** *Let $\pi$ be any policy for the bandits with ranking feedback problem that satisfies and instance-dependent regret upper bound of the form $R_T \leq \sum_{i=1}^{n} C(\Delta_i)f(T)$, where $f(\cdot)$ is a subpolynomial function. Then, $C(\Delta)$ is super-polynomial in $1/\Delta$.*

*Proof.* We prove the opposite implication, namely that if $C(\Delta)$ is polynomial in $1/\Delta$, then the instance-independent regret bound cannot be subpolynomial in $T$. By assumption, in case of two arms with just one gap $\Delta$, we have:

$$R_T \leq \sum_{\ell=1}^{p} C_\ell \Delta^{-\ell} f(T),$$

which implies that the instance-independent regret can be bounded in the following way

$$R_T \leq \sup_{\Delta > 0} \min\left\{\sum_{\ell=1}^{p} C_\ell \Delta^{-\ell} f(T), \Delta T\right\} \leq \sum_{\ell=1}^{p} \sup_{\Delta > 0} \min\left\{C_\ell \Delta^{-\ell} f(T), \Delta T\right\}.$$

Notice that, for $\Delta \geq T^{-1/(\ell+1)}$ the first term is less than $CT^{\frac{\ell}{\alpha+1}}f(T)$, while for $\Delta \leq T^{-1/(\ell+1)}$, the second one is less than $T^{\frac{\ell}{\ell+1}}$. Therefore, the full instance-independent regret is bounded by:

$$R_T \leq \sum_{\ell=1}^{p} C_\ell T^{\frac{\ell}{\ell+1}} f(T),$$

which is polynomial in $T$. If, by contradiction, $f(T)$ were subpolynomial, this bound would be sublinear in $T$, but this contradicts the result of Corollary 2. $\qquad\square$

### 3.4 Instance-independent upper bound

The impossibility result stated in Corollary 2 pushes for the need for an algorithm guaranteeing sublinear regret in the instance-independent case. Initially, we observe that the standard Explore-then-Commit (EC) algorithm [Auer et al., 2002] can be applied within our framework, achieving an $\mathcal{O}(T^{2/3})$ instance-independent regret bound. In the following, we provide a brief overview of how the EC algorithm works. It divides the time horizon into two phases as follows: (i) *exploration phase*: the arms are pulled uniformly for the first $m \cdot n$ rounds,

---

**Algorithm 2** Ranking Logarithmic Phased Elimination (R-LPE)

---
1: initialize $S = [n]$
2: initialize $\mathcal{L} = LG(1/2, 1, T)$
3: **for** $t \in [T]$ **do**
4:     play $i_t \in \arg\min_{i \in S} Z_i(t)$
5:     update $Z_{i_t}(t)$ number of times $i_t$ has been pulled
6:     observe ranking $\mathcal{R}_t$
7:     **if** $\min_{i \in S} Z_i(t) \in \mathcal{L}$ **then**
8:         $\alpha = \frac{\log(\min_{i \in S} Z_i(t))}{\log(T)} - \frac{1}{2}$
9:         $S = \mathcal{F}_t(T^{2\alpha})$
10:   **end if**
11: **end for**

---

where $m$ is a parameter of the algorithm one can tune to minimize the regret; (ii) *commitment phase*: the arm maximizing the estimated reward is pulled. In the case of bandits with ranking feedback, the EC algorithm explores the arms in the first $m \cdot n$ rounds and subsequently pulls the arm in the first position of the ranking feedback received at round $t = m \cdot n$. As is customary in standard (non-ranking) bandits, the best regret bound can be achieved by setting $m = \lceil T^{2/3} \rceil$, thus obtaining an $\mathcal{O}(T^{2/3})$ regret upper bound. We show that we can get a regret bound better than that of the EC algorithm. In particular, we provide the Ranking Logarithmic Phased Elimination (R-LPE) algorithm, which breaks the barrier of $\mathcal{O}(T^{2/3})$ guaranteeing an $\widetilde{\mathcal{O}}(\sqrt{T})$ regret bound when neglecting logarithmic terms. Due to the mathematical instruments involved, the proof of this regret bound only holds for the case of Gaussian rewards, as the ones presented in a similar setting by Garivier et al. [2016]. The pseudocode of R-LPE is reported in Algorithm 2.

In order to proper analyze the algorithm, we need to introduce the two following definitions. Initially, we introduce the definition of the loggrid set as follows.

**Definition 2** (Loggrid). *Given $a, b \in \mathbb{R}$ s.t $a < b$ and a constant value $T > 0$, we define:*

$$LG(a, b, T) := \left\{ \lfloor T^{\lambda_j b + (1-\lambda_j)a} \rfloor : \lambda_j = \frac{j}{\lfloor \log(T) \rfloor}, \ \forall j = 0, \dots, \lfloor \log(T) \rfloor \right\}.$$

Next, we give the notion of active set, which the algorithm employs to cancel out sub-optimal arms.

**Definition 3** (Filtering condition). *Let $S$ be the active set of the algorithm, at a certain timestep. We say that a timestep $t \in [T]$ is **fair** if all active arms have been pulled the same number of times times. In any fair timestep $t \in [T]$, we define the active set $\mathcal{F}_t(\zeta)$ as the set of arms such that:*

$$\mathcal{F}_t(\zeta) := \left\{ i \in S : \forall j \in S \sum_{\tau=1 \text{ s.t. } \tau \text{ fair}}^{t} \{ \mathcal{R}_\tau(i) > \mathcal{R}_\tau(j) \} \geq \zeta \right\}.$$

*This condition will be called **filtering condition**.*

Initially, we observe that R-LPE differs from Algorithm 1, as it takes into account the entire history of the process, not just the most recent ranking $\mathcal{R}_t$. It also requires knowledge of $T$. We denote with $S$ the set of active arms used by the algorithm. Initially, the set $S$ comprises all the possible

arms available in the problem (Line 1). Furthermore, the set which drives the update of the decision space $S$, namely $\mathcal{L}$, is initialized as the loggrid built on parameters $1/2, 1, T$ (Line 2). At every round $t \in [T]$, R-LPE chooses the arm from the active set $S$ with the minimum number of pulls, namely $i \in [n]$ s.t. $Z_i(t)$ is minimized (Line 4); ties are broken by index order. Next, the number of times arm $i_t$ has been pulled, namely $Z_{i_t}(t)$, is updated accordingly (Line 5). The peculiarity of the algorithm is that the set $S$ changes every time the condition $\min_i Z_i(t) \in \mathcal{L}$ is satisfied (Line 7). When the aforementioned condition is met, the set of active arms $S$ is filtered to avoid the exploration on sub-optimal arms. Precisely, $S$ is filtered given the time dependent parameter $\alpha$ (Lines 8- 9). We state the following theorem providing an instance-independent regret bound to Algorithm 2.

**Theorem 4.** *In the stochastic bandits with ranking feedback setting, when the noise is Gaussian, Algorithm 2 achieves $R_T \leq 62n^4 \log(T)^2 T^{1/2}$.*

*Proof.* (Sketch) To prove the theorem, we define, for every pair of indices $i, j \in [n]$, the event:

$$E_{ij}^{\psi} := \begin{cases} E_{ij} & \mu_j - \mu_i > \psi \\ \emptyset & \text{else,} \end{cases} \tag{1}$$

where $E_{ij}$ corresponds to the event in which arm $i$ eliminates arm $j$ at some point in the process, while $\psi$ is a constant defined in the following. The probability that at least one of these events holds is bounded (by Lemma 6 and employing a union bound) as $\mathbb{P}(\Psi) := \mathbb{P}\left(\bigcup_{i \neq j, i, i \in [n]}^{n} E_{ij}^{\psi}\right) \leq \mathcal{O}\left(n^2 \log(T)^2 T^{-1/2} \psi^{-1}\right)$, since their number is at most $n(n-1)/2$.

Therefore, if the complement of the event $\Psi$ holds, there exists an arm $i^\star$ with a gap (w.r.t. the first arm) less than $(n-1)\psi$, which is not eliminated until the last round. This is because at most $n-1$ eliminations can happen, and if the complement of the event $\Psi$ holds, such eliminations concern pairs of arms with a difference in mean of at most $\psi$.

Since the arm $i^\star \in [n]$ is not eliminated until the last round under the event $\Psi^C$, the probability of event $E_{ii^\star}^*$, corresponding to the event in which the suboptimal arm $i \in [n]$ survives for more than $\mathcal{O}(\log(T) T^{1/2} \Delta_{ii^\star}^{-1})$ pulls, is bounded by $2T^{-1/2}$ thanks to Lemma 7. As a result, employing a union bound over all possible values of $i^\star$, we can say that the probability that any event $E_{ii^\star}^*$ with $i^\star \in [n]$ occuring is at most $2(n-1)T^{-1/2}$. Thus, fixing $\psi = \Delta_i/(2(n-1))$ ensures that $\Delta_{ii^\star}^{-1} \geq \Delta_i/n$, which entails $\mathbb{P}(Z_i(T) \geq \mathcal{O}(\log(T) T^{1/2} \Delta_i^{-1})) \leq \mathcal{O}(n^3 \log(T)^2 T^{-1/2} \Delta_i^{-1}))$, and thus, it holds $\mathbb{E}[\Delta_i Z_i(T)] = O(n^3 \log(T)^2 T^{-1/2})$. Finally, using the Regret Decomposition Lemma [Lattimore and Szepesvari, 2017] we can conclude the proof. Proving the two lemmas, however, is not trivial since it requires to see the whole process as a discretization of a biased Brownian motion, and then applying results for this kind of stochastic processes. $\qquad\square$

At first glance, the result presented in Theorem 4 may seem unsurprising. Indeed, there are several elimination algorithms achieving $\mathcal{O}(\sqrt{T})$ regret bounds in different bandit settings (see, for example, [Auer and Ortner, 2010, Lattimore et al., 2020, Li and Scarlett, 2022]). Nevertheless, our setting poses several additional challenges compared to existing ones. For instance, in our framework, it is not possible to rely on concentration bounds, as the current feedback is heavily correlated with the past ones. This is precisely the reason for the anomalous growth of the regret in terms of the number of arms $n$: due to the extremely correlated feedback, two union bounds are necessary trough the last proof, and this leads to an increase in the dependence of the order of $n$. In the impossibility of using concentration inequalities, our analysis employs novel arguments, drawing from recent results in the theory of Brownian Motions, which allow to properly model the particular ranking feedback.

## 4 Analysis in the adversarial setting

We focus on bandits with ranking feedback in adversarial settings. In particular, we show that no algorithm provides sublinear regret without statistical assumptions on the rewards.

**Theorem 5.** *In adversarial bandits with ranking feedback, there exists a constant $\gamma \in (0,1)$ such that no algorithm achieves $o(T)$ regret with respect to the best arm in hindsight with probability greater than $\gamma$.*

*Proof.* (Sketch) The proof introduces three instances in an adversarial setting in a way that no algorithm can achieve sublinear regret in all three. The main reason behind such a negative result is that ranking feedback obfuscates the value of the rewards so as not to allow the algorithm to distinguish two or more instances where the rewards are non-stationary. The three instances employed in the proof are divided into three phases such that the instances are similar in terms of rewards for the first two phases, while they are extremely different in the third phase. In summary, if the learner receives the same ranking when playing in two instances with different best arms in hindsight, it is not possible to achieve a small regret in both of them. □

## Acknowledgments

This paper is supported by the FAIR (Future Artificial Intelligence Research) project, funded by the NextGenerationEU program within the PNRR-PE-AI scheme (M4C2, Investment 1.3, Line on Artificial Intelligence), by the EU Horizon project ELIAS (European Lighthouse of AI for Sustainability, No. 101120237) and by project SERICS (PE00000014) under the NRRP MUR program funded by the EU - NGEU.

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

# A  Proofs of instance dependent stochastic analysis

## A.1  Proof of instance dependent lower bound and lemmas

**Lemma 1** (Separation lemma)**.** *Let $\{G_t\}_{t\in\mathbb{N}}, \{G'_t\}_{t\in\mathbb{N}}$ be two independent random walks defined as:*

$$G_{t+1} = G_t + \epsilon_t \qquad and \qquad G'_{t+1} = G'_t + \epsilon'_t,$$

*where $G_0 = G'_0 = 0$ and the drifts satisfy $\mathbb{E}[\epsilon_t] = p > q = \mathbb{E}[\epsilon'_t]$, for each $t \in \mathbb{N}$. Then, we have:*

$$\mathbb{P}\Big(\forall t, t' \in \mathbb{N}^* \ \ G_t/t \geq G'_{t'}/t'\Big) \geq c(p,q) > 0.$$

*Proof.* Let us consider the random walk defined as follows:

$$\widetilde{G}_{t+1} = \widetilde{G}_t + \epsilon_t - \frac{p+q}{2}.$$

Since $\mathbb{E}\left[\epsilon_t - \frac{p+q}{2}\right] > 0$, and observing that a random walk with a non null drift is transient, we know that there exists a constant $c_1(p,q) > 0$ such that:

$$\mathbb{P}\Big(\widetilde{G}_t > 0, \ \forall t > 0\Big) = c_1(p,q) > 0.$$

On the other hand, we observe that the random walk:

$$\widetilde{G}'_{t+1} = \widetilde{G}'_t + \epsilon'_t - \frac{p+q}{2}$$

satisfies the opposite inequality since $\mathbb{E}[\epsilon'_t - \frac{p+q}{2}] < 0$. Therefore, for the same reason as above, we have:

$$\mathbb{P}\Big(\widetilde{G}'_t < 0, \ \forall t > 0\Big) = c_2(p,q) > 0,$$

for some constant $c_2(p,q) > 0$.

Thus, since the two processes are independent, we have:

$$\mathbb{P}\Big(\bigcap_{t,t'=1}^{\infty} \{\widetilde{G}_{t+1} > 0, \widetilde{G}'_{t'+1} < 0\}\Big) > 0,$$

which entails:

$$
\begin{aligned}
c_1(p,q)c_2(p,q) &= \mathbb{P}\Big(\bigcap_{t,t'=1}^{\infty} \{\widetilde{G}_t > 0, \widetilde{G}'_{t'} < 0\}\Big) \\
&= \mathbb{P}\Big(\bigcap_{t,t'=1}^{\infty} \{G_t - t\frac{p+q}{2} > 0, G'_{t'} - t'\frac{p+q}{2} < 0\}\Big) \\
&= \mathbb{P}\Big(\bigcap_{t,t'=1}^{\infty} \{G_t/t > \frac{p+q}{2}, G'_{t'}/t' < \frac{p+q}{2}\}\Big) \\
&\leq \mathbb{P}\Big(\bigcap_{t,t'=1}^{\infty} \{G_t/t > G'_{t'}/t'\}\Big).
\end{aligned}
$$

The latter inequality can be easily reformulated as in the statement of the lemma, concluding the proof. $\square$

In order to prove the lower bound, we will also show the following lemma.

**Lemma 2.** *Let $\{X_i\}_{i\in[n]}$ be a sequence of i.i.d. Bernoulli random variables. Then, for every event $E \in \mathcal{F}_n$, where $\mathcal{F}_n$ is the filtration generated by $X_1, \ldots X_n$, we have:*

$$\mathbb{P}_{X_1,\ldots X_n \sim Be(p)}(E) \leq \mathbb{P}_{X_1,\ldots X_n \sim Be(p')}(E) \max\left(\frac{p}{p'}, \frac{1-p}{1-p'}\right)^n$$

*Proof.* Without loss of generality, let us assume that $p' < p$. Then, we have:

$$\mathbb{P}_{X_1,\ldots X_n \sim Be(p)}(E) = \int_{\{0,1\}^n} \mathbf{1}_E(\underline{x}) \prod_{i=1}^n p^{x_i}(1-p)^{1-x_i} d\underline{x}$$

$$\leq \int_{\{0,1\}^n} \mathbf{1}_E(\underline{x}) \prod_{i=1}^n (p/p')^{x_i} p'^{x_i}(1-p')^{1-x_i} d\underline{x} \qquad (2)$$

$$\leq \left(\frac{p}{p'}\right)^n \int_{\{0,1\}^n} \mathbf{1}_E(\underline{x}) \prod_{i=1}^n p'^{x_i}(1-p')^{1-x_i} d\underline{x}$$

$$= \left(\frac{p}{p'}\right)^n \mathbb{P}_{X_1,\ldots X_n \sim Be(p')}(E).$$

where Equation (2) follows from the assumption that $p' < p$. The other way round can be proved substituting $p$ and $p'$. This concludes the proof. $\qquad \square$

**Theorem 1** (Instance-dependent lower bound). *Let $\pi$ be any policy for the bandits with ranking feedback problem, then, for any $C : [0,+\infty) \to [0,+\infty)$, there exists a $\Delta > 0$ and a time horizon $T > 0$ such that $R_T > C(\Delta)\log(T)$.*

*Proof.* Let $p_1 = 0.5$, $p_2 = 0.5 - \Delta$, $p_2^* = 0.5 + \Delta$, for some $0 < \Delta \leq 1/4$ (independent on $T$) that satisfies:

$$2C(1)\log(1 - 4\Delta) \geq -1/2.$$

We consider the following three instances:

$$\mathcal{P}: \begin{cases} \nu_1 = Be(p_1) \\ \nu_2 = Be(p_2) \end{cases} \qquad \mathcal{P}^*: \begin{cases} \nu_1 = Be(p_1) \\ \nu_2 = Be(p_2^*) \end{cases} \qquad \mathcal{P}^{**}: \begin{cases} \nu_1 = Be(1) \\ \nu_2 = Be(0) \end{cases}$$

Clearly, the optimal arm is the first in instances $\mathcal{P}$ and $\mathcal{P}^{**}$, while the optimal arm is the second in instance $\mathcal{P}^*$. By contradiction, we assume that, for every time horizon $T > 0$, exists a policy $\pi_T(\cdot|H_t)$ for each $t < T$ satisfying, in all three cases, the following condition:

$$R_T(\pi_T) \leq C(\Delta)\log(T). \qquad (3)$$

We also define the following event about the rankings received by the learner:

$$E_t = \bigcap_{\tau=1}^t \{\mathcal{R}_\tau = \langle 1, 2\rangle\}.$$

The event $E_t$ can be interpreted as "up to time $t$, the learner has always observed the ranking $\langle 1, 2\rangle$". Note that by applying Equation 3 to the particular case of instance $\mathcal{P}^{**}$, we have:

$$C(1)\log(T) \geq R_T(\pi_T|\mathcal{P}^{**}) = \mathbb{E}_{\mathcal{P}^{**}}[Z_2(T)] = \mathbb{E}_{\mathcal{P}^{**}}[Z_2(T)|E_T],$$

where the last step holds since in instance $\mathcal{P}^{**}$ the event $E_T$ holds almost surely. Therefore, we have:

$$\mathbb{E}_{\mathcal{P}^{**}}[Z_2(T)|E_T] \leq C(1)\log(T).$$

Let $h \in \mathbb{N}$. Then, we have:

$$\mathbb{P}_{\mathcal{P}}(Z_2(T) < h|E_T) = \mathbb{P}_{\mathcal{P}^{**}}(Z_2(T) < h|E_T)$$

$$\geq 1 - \frac{C(1)\log(T)}{h}.$$

We note that the above equality holds since, under the event $E_T$, no policy can distinguish between the two instances $\mathcal{P}^{**}$ and $\mathcal{P}$. On the other hand, the above inequality follows from Markov's theorem.

Furthermore, we notice that the event $\{Z_2(T) < h\}$ is contained in the $\sigma-$algebra generated by the first $h$ pulls of arm 2 (and all the pulls of arm 1, but this is irrelevant since $\nu_1$ corresponds to the same distribution in the first two instances). Therefore, for all $h > 0$, from Lemma 2 we have:

$$\mathbb{P}_{\mathcal{P}^*}(Z_2(T) < h) \geq \left(\frac{0.5 - \Delta}{0.5 + \Delta}\right)^h \mathbb{P}_{\mathcal{P}}(Z_2(T) < h)$$

$$\geq \left(\frac{0.5 - \Delta}{0.5 + \Delta}\right)^h \mathbb{P}_{\mathcal{P}}(Z_2(T) < h, E_T)$$

$$= \left(\frac{0.5 - \Delta}{0.5 + \Delta}\right)^h \mathbb{P}_{\mathcal{P}}(Z_2(T) < h | E_T) \mathbb{P}_{\mathcal{P}}(E_T)$$

$$\geq \left(\frac{0.5 - \Delta}{0.5 + \Delta}\right)^h \left(1 - \frac{C(1)\log(T)}{h}\right) \kappa(\Delta)$$

Where $\kappa(\Delta) := \inf_{t>0} \mathbb{P}_P(E_t) > 0$ thanks to Lemma 1.

Here, for every $1/4 \geq x > 0$, the inequality $0.5 - x \geq (1 - 4x)(0.5 + x) \geq 0$ holds. Thus, for all $h > 0$, we have:
$$\left(\frac{0.5 - \Delta}{0.5 + \Delta}\right)^h \geq \left(1 - 4\Delta\right)^h.$$

Therefore, taking $h = 2C(1)\log(T)$, we have:

$$\mathbb{P}_{\mathcal{P}^*}(Z_2(T) < h) \geq \left(\frac{0.5 - \Delta}{0.5 + \Delta}\right)^h \left(1 - \frac{C(1)\log(T)}{h}\right)\kappa(\Delta)$$

$$\geq \frac{\kappa(\Delta)}{2}\left(1 - 4\Delta\right)^{2C(1)\log(T)}$$

$$= \frac{\kappa(\Delta)}{2}T^{2C(1)\log(1-4\Delta)}.$$

Thus, we can lower bound on the regret in case of instance $\mathcal{P}^*$ as follows:

$$R_T(\pi_T|\mathcal{P}^*) \geq \Delta \mathbb{E}_{\mathcal{P}^*}[(T - Z_2(T))]$$

$$\geq \Delta(T - h)\mathbb{P}_{\mathcal{P}^*}(Z_2(T) < h)$$

$$= \Delta(T - 2C(1)\log(T))\mathbb{P}_{\mathcal{P}^*}(Z_2(T) < 2C(1)\log(T))$$

$$\geq \frac{\kappa(\Delta)\Delta(T - 2C(1)\log(T))}{2}T^{2C(1)\log(1-4\Delta)}$$

$$\geq \frac{\kappa(\Delta)\Delta(T - 2C(1)\log(T))}{2}T^{-1/2},$$

which grows polynomially with time. At this point, Equation (3) would give, for every $T > 0$, the following:

$$C(\Delta)\log(T) \geq \frac{\kappa(\Delta)\Delta(T - 2C(1)\log(T))}{2}T^{-1/2} \geq \Omega\left(\frac{\kappa(\Delta)\Delta}{2}T^{1/2}\right).$$

Clearly, for $T$ sufficiently big, the above statement is false, concluding the proof. $\qquad\square$

### A.2   Proof of instance dependent upper bound

**Theorem 2** (Instance-dependent upper bound). *Assume that the reward distribution of every arm is 1-subgaussian. Let $f : (0, \infty) \to \mathbb{R}_+$ be a superlogarithmic nondecreasing function in t. Then there is a term $C(f, \Delta_i)$ for each sub-optimal arm $i \in [n]$ which does not depend on T, such that Algorithm 1 satisfies $R_T \leq (1 + f(T)) \sum_{i=1}^n \Delta_i + \log(T) \sum_{i=1}^n C(f, \Delta_i)$.*

*Proof.* Let $i^* \in [n]$ be the optimal arm. For any suboptimal arm $i \in [n]$, we have:

$$\mathbb{E}[Z_i(T)] = \sum_{\tau=1}^{T} \mathbb{P}(i_\tau = i)$$

$$\leq 1 + \sum_{\tau=n+1}^{T} \mathbb{P}(i_\tau = i, Z_i(\tau - 1) < f(\tau)) + \sum_{\tau=n+1}^{T} \mathbb{P}(i_\tau = i, Z_i(\tau - 1) \geq f(\tau)), \quad (4)$$

where we let $i_\tau \in [n]$ be the arm pulled at time $\tau \in [T]$. We split the proof in two parts, providing a bound for each term defining Equation (4).

**Claim 1:** The first term of Equation (4) is bounded by $f(T)$. Indeed, notice that, being $f(\cdot)$ non-decreasing, we have:

$$\sum_{\tau=1}^{T} \mathbf{1}\{i_\tau = i, Z_i(\tau - 1) < f(\tau)\} \leq \sum_{\tau=1}^{T} \sum_{\rho=1}^{f(\tau)} \mathbf{1}\{i_\tau = i, Z_i(\tau - 1) = \rho\}$$

$$\leq \sum_{\tau=1}^{T} \sum_{\rho=1}^{f(T)} \mathbf{1}\{i_\tau = i, Z_i(\tau - 1) = \rho\}$$

$$= \sum_{\rho=1}^{f(T)} \sum_{\tau=1}^{T} \mathbf{1}\{i_\tau = i, Z_i(\tau - 1) = \rho\} \leq f(T).$$

**Claim 2:** The second term in Equation (4) is bounded by $C(\Delta_i) \log(T)$ for some $C(\Delta_i)$. We know, by design of the algorithm, that the arm $i \in [n]$ can be pulled only if:

1. It has the highest empirical mean.

2. Every other arm has been pulled at least $f(T)$ times, including arm $i^* \in [n]$.

We define the event:
$$E_{i,t} := \{Z_i(t) \geq f(t)\}.$$

Then, we have:
$$\mathbb{P}(i_\tau = i, Z_i(\tau - 1) \geq f(\tau)) \leq \mathbb{P}(\hat{r}_\tau(i) > \hat{r}_\tau(i^*), E_{i,t}, E_{i^*,t})$$

which can be true only if at least one of the following holds:

1. $\hat{r}_\tau(i) > \mu_i + \Delta_i/2$, which, when intersected with $E_{i,\tau}$, holds thanks Hoeffding's inequality with probability at most:

$$\mathbb{P}(\hat{r}_\tau(i) > \mu_i + \Delta_i/2, E_{i,\tau}) \leq \sum_{y \geq f(\tau)}^{\infty} \mathbb{P}(\hat{r}_\tau(i) > \mu_i + \Delta_i/2, Z_i(\tau) = y)$$

$$\leq \sum_{y \geq f(\tau)}^{\infty} e^{-\frac{y\Delta_i^2}{2}} = \frac{e^{-\frac{f(\tau)\Delta_i^2}{2}}}{1 - e^{-\Delta_i^2/2}}$$

2. $\hat{r}_\tau(i^*) < \mu_{i^*} - \Delta_i/2$, which, when intersected to $E_{i^*,\tau}$, is also true with the same probability as above, thanks to Hoeffding's inequality.

Therefore, we have proved that:

$$\sum_{\tau=1}^{T} \mathbb{P}(i_\tau = i, Z_i(\tau - 1) \geq f(\tau)) \leq 2(1 - e^{-\Delta_i^2/2})^{-1} \sum_{\tau=1}^{T} e^{-\frac{f(\tau)\Delta_i^2}{2}}$$

$$\leq \underbrace{2(1 - e^{-\Delta_i^2/2})^{-1} \frac{\sum_{\tau=1}^{T} e^{-\frac{f(\tau)\Delta_i^2}{2}}}{\log(T)}}_{(*)} \log(T).$$

We show that there exists a constant $C_0(\Delta_i) < +\infty$ such that $(*) \leq C_0(\Delta_i)$ in the equation above. We start observing that, since $f(\cdot)$ is superlogathmic, we have:

$$\limsup_{t\to\infty} t e^{-\frac{f(t)\Delta_i^2}{2}} = \limsup_{t\to\infty} t e^{-\log(t)\frac{f(t)\Delta_i^2}{2\log(t)}} = \limsup_{t\to\infty} t \left(\frac{1}{t}\right)^{\frac{f(t)\Delta_i^2}{2\log(t)}} = 0.$$

Thus, we can find $t_0$ such that $e^{-\frac{f(t)\Delta_i^2}{2}} \leq \frac{1}{t}$ for all $t \geq t_0$, so that:

$$\limsup_{t\to\infty} \frac{\sum_{\tau=1}^t e^{-\frac{f(\tau)\Delta_i^2}{2}}}{\log(t)} \leq \limsup_{t\to\infty} \frac{\sum_{\tau=1}^t e^{-\frac{f(\tau)\Delta_i^2}{2}}}{\sum_{\tau=1}^t \frac{1}{\tau}} \leq \limsup_{t\to\infty} \underbrace{\frac{\sum_{\tau=1}^{t_0} e^{-\frac{f(\tau)\Delta_i^2}{2}}}{\sum_{\tau=1}^t \frac{1}{\tau}}}_{\to 0} + \underbrace{\frac{\sum_{\tau=t_0}^t e^{-\frac{f(\tau)\Delta_i^2}{2}}}{\sum_{\tau=t_0}^t \frac{1}{\tau}}}_{\leq 1} \leq 1.$$

Thus, since a sequence with finite limit superior is always bounded, we have:

$$C_0(\Delta_i) := 2(1 - e^{-\Delta_i^2/2})^{-1} \sup_{t>1} \frac{\sum_{\tau=1}^t e^{-\frac{f(\tau)\Delta_i^2}{2}}}{\log(t)} < +\infty,$$

proving that for every suboptimal arm $i \in [n]$ the following holds:

$$\mathbb{E}[Z_i(T)] \leq 1 + f(T) + C_0(\Delta_i) \log(T).$$

Finally, defining $C(\Delta_i) := \Delta_i C_0(\Delta_i)$, we have:

$$R_T = \sum_{i=1}^n \Delta_i \mathbb{E}[Z_i(T)],$$

concluding the proof. □

**Corollary 1.** *Let* $\delta > 0$ *and* $f(t) = \log(t)^{1+\delta}$ *be the sperlogarithmic function used in Algorithm 1, then we have:*

$$C(f, \Delta_i) = \frac{2\Delta_i \left(e^{\left((2/\Delta_i^2)^{1/\delta}\right)} + 1\right)}{1 - e^{-\Delta_i^2/2}}.$$

*Proof.* Let $t_0 > 1$ be smallest integer such that:

$$e^{-\frac{\log(t)^{1+\delta}\Delta_i^2}{2}} \leq \frac{1}{t},$$

for all $t \geq t_0 > 1$. By rearranging the latter inequality we have that:

$$t_0 = \lceil e^{\left((2/\Delta_i^2)^{1/\delta}\right)} \rceil.$$

From the proof of Theorem 2, we have $C(\Delta_i, f) := \Delta_i C_0(\Delta_i)$, where:

$$C_0(\Delta_i) = 2(1 - e^{-\Delta_i^2/2})^{-1} \sup_{t>1} \frac{\sum_{\tau=1}^t e^{-\frac{f(\tau)\Delta_i^2}{2}}}{\log(t)}.$$

Furthermore, we let $t_\star > 1$ be the integer satisfying:

$$\frac{\sum_{\tau=1}^t e^{-\frac{f(\tau)\Delta_i^2}{2}}}{\log(t)} \leq \frac{\sum_{\tau=1}^{t_\star} e^{-\frac{f(\tau)\Delta_i^2}{2}}}{\log(t_\star)},$$

for all the integers $t > 1$. Notice that such a value always exists, as the supremum limit of the above quantity is finite. Therefore, we have:

$$C_0(\Delta_i) \log(t_\star) = 2(1 - e^{-\Delta_i^2/2})^{-1} \sum_{\tau=1}^{t_\star} e^{-\frac{f(\tau)\Delta_i^2}{2}}$$

$$= 2(1 - e^{-\Delta_i^2/2})^{-1} \frac{\sum_{\tau=1}^{t_\star} e^{-\frac{f(\tau)\Delta_i^2}{2}}}{\log(t_\star)} \log(t_\star)$$

$$\leq 2(1 - e^{-\Delta_i^2/2})^{-1} \left( \frac{\sum_{\tau=1}^{t_0} e^{-\frac{f(\tau)\Delta_i^2}{2}}}{\log(t_\star)} + \frac{\sum_{\tau=t_0}^{t_\star} e^{-\frac{f(\tau)\Delta_i^2}{2}}}{\sum_{\tau=t_0}^{t_\star} \frac{1}{\tau}} \right) \log(t_\star)$$

Where the last step is due to the fact that for $t_0 > 1$ we have $\sum_{\tau=t_0}^{t_\star} \frac{1}{\tau} \le \log(t_\star)$.

Furthermore, thanks to the definition of $t_0$, we have:

$$\frac{\sum_{\tau=t_0}^{t_\star} e^{-\frac{f(\tau)\Delta_i^2}{2}}}{\sum_{\tau=t_0}^{t_\star} \frac{1}{\tau}} \le 1.$$

With this consideration, we are able to conclude:

$$C_0(\Delta_i) \le 2(1 - e^{-\Delta_i^2/2})^{-1} \left( \frac{\sum_{\tau=1}^{t_0} e^{-\frac{f(\tau)\Delta_i^2}{2}}}{\log(t_\star)} + \frac{\sum_{\tau=t_0}^{t_\star} e^{-\frac{f(\tau)\Delta_i^2}{2}}}{\sum_{\tau=t_0}^{t_\star} \frac{1}{\tau}} \right)$$

$$\le 2(1 - e^{-\Delta_i^2/2})^{-1} \left( \frac{t_0}{\log(t_\star)} + 1 \right)$$

$$\le 2(1 - e^{-\Delta_i^2/2})^{-1} + 2(1 - e^{-\Delta_i^2/2})^{-1} e^{\left( (2/\Delta_i^2)^{1/\delta} \right)}.$$

Where in the last step we have used the fact that $\log(t) > 1$ for $t > 2$. Recollecting all the terms we have:

$$C(\Delta_i, \log(t)^{1+\delta}) = \Delta_i C_0(\Delta_i, \log(t)^{1+\delta}) \le 2\Delta_i (1 - e^{-\Delta_i^2/2})^{-1} (e^{\left( (2/\Delta_i^2)^{1/\delta} \right)} + 1),$$

concluding the proof. $\qquad\square$

# B   Proofs in the instance independent stochastic analysis

## B.1   Instance dependent/independent trade-off

**Lemma 3.** *Let us define a random walk*

$$G_{t+1} = G_t + \epsilon_t \qquad \epsilon_t = \begin{cases} 1 & \text{with prob. } p \\ -1 & \text{with prob. } 1-p \end{cases}.$$

*with $G_0 = 1$ and $p = 1/2 + \Delta/2 > 0.5$ for some $\Delta \in (0,1)$. Then, we have:*

$$\mathbb{P}\left( \bigcup_{t=1}^{\infty} \{G_t \le 0\} \right) = \left( \frac{1-\Delta}{1+\Delta} \right).$$

*Proof.* We define:

$$f_n = \mathbb{P}(G_0 = n, \exists t : G_t = 0)$$

which satisfies, for $n \ge 0$, the following recursive equation:

$$f_n = p f_{n-1} + (1-p) f_{n+1}$$

with $f_n = 1$ for $n \le 0$. The equation corresponding to the aforementioned dynamical system is:

$$(1-p)\lambda^2 - \lambda + p = 0.$$

The two solutions of the above equation are:

$$\lambda_{1,2} = \frac{1 \pm \sqrt{1 - 4p(1-p)}}{2(1-p)}$$

Thus, for all $n > 0$, we obtain:

$$f_n = A \left( \frac{1 + \sqrt{1 - 4p(1-p)}}{2(1-p)} \right)^n + B \left( \frac{1 - \sqrt{1 - 4p(1-p)}}{2(1-p)} \right)^n,$$

where $A = 0$ (otherwise, the equation does not define a probability) and $B = 1$ (since $f_1 \to 1$ for $\Delta \to 0$). Therefore, from the definition of $p$, we get:

$$f_n = \left( \frac{1 - \sqrt{1 - 4p(1-p)}}{2(1-p)} \right)^n$$

$$= \left( \frac{1 - \sqrt{1 - 4(1/2 - \Delta/2)(1/2 + \Delta/2)}}{1 + \Delta} \right)^n$$

$$= \left( \frac{1 - \Delta}{1 + \Delta} \right)^n.$$

The lemma holds by setting $n = 1$. $\qquad\qquad\qquad\qquad\qquad\qquad\qquad\square$

**Theorem 3** (Instance Dependent/Independent Trade-off). *Let $\pi$ be any policy for the bandits with ranking feedback problem. If $\pi$ satisfies the following properties:*

- *(instance-dependent regret upper bound) $R_T \leq \sum_{i=1}^n C(\Delta_i) T^\alpha$*

- *(instance-independent regret upper bound) $R_T \leq nCT^\beta$*

*then, $2\alpha + \beta \geq 1$, where $\alpha, \beta \geq 0$.*

*Proof.* Let $p_1 = 0.5$, $p_2 = 0.5 - \Delta$, $p_2^* = 0.5 + \Delta$. Let us consider three problems:

$$P : \begin{cases} \nu_1 = Be(p_1) \\ \nu_2 = Be(p_2) \end{cases} \qquad P^* : \begin{cases} \nu_1 = Be(p_1) \\ \nu_2 = Be(p_2^*) \end{cases} \qquad P^{**} : \begin{cases} \nu_1 = Be(1) \\ \nu_2 = Be(0) \end{cases}$$

Clearly, the optimal arm is the first in instances $\mathcal{P}$ and $\mathcal{P}^{**}$, while the optimal arm is the second in instance $\mathcal{P}^*$.

Let us now define the event:

$$E_t = \bigcap_{\tau=1}^{t} \{ \mathcal{R}_t = \langle 1, 2 \rangle \}.$$

By assumption, the policy $\pi$ has a sub-$T^\alpha$ instance-dependent regret. Therefore, in instance $\mathcal{P}^{**}$, we have for all $\eta > 0$ the following:

$$\limsup_{T \to \infty} \frac{\mathbb{E}_{\mathcal{P}^{**}}[Z_2(T)|E_T]}{T^{\alpha+\eta}} = \limsup_{T \to \infty} \frac{\mathbb{E}_{\mathcal{P}^{**}}[Z_2(T)]}{T^{\alpha+\eta}} = \limsup_{T \to \infty} \frac{R_T}{T^{\alpha+\eta}} = 0,$$

since in the instance $\mathcal{P}^{**}$, the event $E_T$ holds almost surely.

Under this event, no policy can distinguish between the instances. Therefore,

$$\limsup_{t \to \infty} \frac{\mathbb{E}_{\mathcal{P}}[Z_2(t)|E_t]}{T^{\alpha+\eta}} = \limsup_{T \to \infty} \frac{\mathbb{E}_{\mathcal{P}^{**}}[Z_2(t)|E_t]}{T^{\alpha+\eta}} = 0$$

$$\implies \exists C > 0 \; \forall t > 0 \quad : \quad \mathbb{E}_{\mathcal{P}}[Z_2(t)|E_t] \leq Ct^{\alpha+\eta}.$$

Where $\mathbb{E}_{\mathcal{P}}[\cdot]$ is the expectation over the random variables of the rewards in instance $\mathcal{P}$. Therefore, by Markov's inequality, we have:

$$\mathbb{P}_{\mathcal{P}}(Z_2(T) > 2CT^{\alpha+\eta}|E_T) \leq \frac{\mathbb{E}_{\mathcal{P}}[Z_2(T)|E_T]}{2CT^{\alpha+\eta}} \leq \frac{CT^{\alpha+\eta}}{2CT^{\alpha+\eta}} \leq \frac{1}{2}.$$

Now, note that for every $h > 0$, the event $\{Z_2(T) < h\}$ is contained in the $\sigma-$algebra generated by the first $h$ pulls of arm 2 (and all the pulls of arm 1, but this is irrelevant since arm 1 corresponds to the same distribution in the first two instances). Therefore, thanks to Lemma 2, we have:

$$\forall h > 0 \qquad \mathbb{P}_{\mathcal{P}^*}(Z_2(T) \leq h) \geq \left( \frac{0.5 - \Delta}{0.5 + \Delta} \right)^h \mathbb{P}_{\mathcal{P}}(Z_2(T) \leq h)$$

$$\geq \left( \frac{0.5 - \Delta}{0.5 + \Delta} \right)^h \mathbb{P}_{\mathcal{P}}(Z_2(T) \leq h, E_T)$$

$$= \left( \frac{0.5 - \Delta}{0.5 + \Delta} \right)^h \mathbb{P}_{\mathcal{P}}(Z_2(T) \leq h|E_T)\mathbb{P}_{\mathcal{P}}(E_T).$$

By the previous step, we have $\mathbb{P}_{\mathcal{P}}(Z_2(T) \leq 2CT^{\alpha+\eta}|E_T) \geq 1/2$, so that:

$$\mathbb{P}_{\mathcal{P}^*}(Z_2(T) \leq 2CT^{\alpha+\eta}) \geq \frac{1}{2} \left( \frac{0.5 - \Delta}{0.5 + \Delta} \right)^{2CT^{\alpha+\eta}} \mathbb{P}_P(E_T),$$

while, thanks to Lemma 3, we have:

$$\mathbb{P}_{\mathcal{P}}(E_T) \geq 1 - \frac{1 - \Delta}{1 + \Delta} \geq 2\Delta,$$

meaning that:

$$\mathbb{P}_{\mathcal{P}^*}(Z_2(T) \leq 2CT^{\alpha+\eta}) \geq \frac{1}{2}\left(\frac{0.5 - \Delta}{0.5 + \Delta}\right)^{2CT^{\alpha+\eta}} \frac{2\Delta}{1 + \Delta}.$$

We use this result to provide a lower bound for the regret in the instance-independent case. Analyzing the instance independent-regret, by definition, we have to fix $T$ as time horizon and let the arm gap $\Delta$ my depend on $T$.

Let us now fix $\rho > 1$. With the choice $\Delta = T^{-\rho\alpha}(\leq 1/4$ for sufficiently big $T$), we have:

$$\begin{aligned}
\mathbb{P}_{\mathcal{P}^*}(Z_2(T) \leq 2CT^{\alpha+\eta}) &\geq \frac{1}{2}\left(\frac{0.5 - T^{-\rho\alpha}}{0.5 + T^{-\rho\alpha}}\right)^{2CT^{\alpha+\eta}} 2T^{-\rho\alpha} \\
&\geq \frac{1}{2}\left(1 - 4T^{-\rho\alpha}\right)^{2CT^{\alpha+\eta}} \frac{2T^{-\rho\alpha}}{1 + T^{-\rho\alpha}} \\
&\geq \left(1 - 4T^{-\rho\alpha}\right)^{2CT^{\alpha+\eta}} \frac{T^{-\rho\alpha}}{2}.
\end{aligned} \tag{5}$$

At this point if we choose $\eta = \alpha(\rho - 1)/2$, the following fact holds:

$$\begin{aligned}
\lim_{T \to \infty} \left(1 - 4T^{-\rho\alpha}\right)^{2CT^{\alpha+\eta}} &= \lim_{y \to 0^+} \left(1 - 4y\right)^{Cy^{\frac{-(\alpha+\eta)}{\rho\alpha}}} \\
&= \lim_{y \to 0^+} \left(1 - 4y\right)^{Cy^{\frac{-\alpha(1/2+\rho/2)}{\rho\alpha}}} \\
&= \lim_{y \to 0^+} \left(1 - 4y\right)^{Cy^{\frac{-(1/2+\rho/2)}{\rho}}}
\end{aligned}$$

where in the first equality we substituted $y = T^{-\rho\alpha}$. Here, $y^{\frac{-(1/2+\rho/2)}{\rho}} = y^{-1} \cdot y^{\frac{\rho-1}{2\rho}}$, where the second exponent is strictly positive.

$$\begin{aligned}
\lim_{T \to \infty} \left(1 - 4T^{-\rho\alpha}\right)^{2CT^{\alpha+\eta}} &= \lim_{y \to 0^+} \left(\left(1 - 4y\right)^{-1/y}\right)^{Cy^{\frac{\rho-1}{2\rho}}} \\
&= \lim_{y \to 0^+} (1/e^4)^{Cy^{\frac{\rho-1}{2\rho}}} = 1.
\end{aligned}$$

This limit shows that there is $c_\rho > 0$ such that:

$$\left(1 - 4T^{-\rho\alpha}\right)^{2CT^{\alpha+\eta}} \geq c_\rho \qquad,$$

when $T > 0$ is sufficiently large.

Substituting this property in Equation (5), we get:

$$\mathbb{P}_{\mathcal{P}^*}(Z_2(T) \leq 2CT^{\alpha(1/2+\rho/2)}) \geq \frac{c_\rho}{2}T^{-\rho\alpha},$$

holding for every $\rho > 0$ and sufficiently large $T > 0$.

Thus, with $\Delta = T^{-\rho\alpha}$, we have:

$$\begin{aligned}
\mathbb{E}_{\mathcal{P}^*}[R_T] &\geq \Delta(T - 2CT^{\alpha(1/2+\rho/2)})\mathbb{P}_{\mathcal{P}^*}(Z_2(T) \leq 2CT^{\alpha(1/2+\rho/2)}) \\
&\geq \frac{1}{2}T \cdot \frac{c_\rho}{2}T^{-2\rho\alpha} = \frac{c_\rho}{4}T^{1-2\rho\alpha}
\end{aligned}$$

for all $T > 0$ sufficiently big.

Therefore, for $\beta \leq 1 - 2\rho\alpha$ it is not possible to have an instance-independent upper regret bound. Since this is valid for every $\rho > 1$, we can also extend the result to any $\beta < 1 - 2\alpha$, which leads to the conclusion that the necessary condition to satisfy both:

- (instance-dependent regret bound)

$$R_T \leq \sum_{i=1}^n C(\Delta_i) T^\alpha \qquad \forall T > 0$$

- (instance-independent regret bound)

$$R_T \leq nCT^\beta \qquad \forall T > 0$$

for the same policy $\pi$ is:

$$2\alpha + \beta \geq 1.$$

$\square$

## B.2  Proofs of instance independent regret upper bound

To derive the final instance-independent regret bound, we introduce some results from the theory of stochastic processes. The following subsections are therefore devoted to developing all the necessary results to prove the final regret bound of the algorithm.

### B.2.1  Discretizing the Brownian motion

In this section, we prove some results about the relationship between random walks and Brownian motions, that will be crucial in the proof of the regret bound. For this scope, we will introduce this quantity:

$$|B_t + t\mu_0 < \eta| = \int_0^1 \mathbf{1}_{(-\infty,\eta)}(\tau) d\tau,$$

corresponding to the Lebesgue measure of the set $\{t \in [0,1] : B_t + t\mu_0 < \eta\}$.

We start with a lemma that bounds the increments in a standard Brownian motion.

**Lemma 4.** *Let $\{B_t\}_{t\in[0,1]}$ be a standard Brownian motion. We define:*

$$I_i := [i/n, (i+1)/n],$$

*for all $i \in \{0, \ldots, n-1\}$. Then, for every $\eta \geq 0$, we have:*

$$\mathbb{P}\left( \sup_{i\in\{0,\ldots,n-1\}} (\sup_{t\in I_i} B_t - B_{i/n}) \geq \eta \right) = \mathbb{P}\left( \inf_{i\in\{0,\ldots,n-1\}} (\inf_{t\in I_i} B_t - B_{i/n}) \leq -\eta \right)$$

$$\leq \frac{2\sqrt{n}\exp\left(-\frac{\eta^2 n}{2\sigma^2}\right)}{\eta/\sigma\sqrt{2\pi}}.$$

*Proof.* We notice that the Brownian motion satisfies:

$$\mathbb{P}\left( \sup_{i\in\{0,\ldots,n-1\}} (\sup_{t\in I_i} B_t - B_{i/n}) \geq \eta \right) = \mathbb{P}\left( \bigcup_{i=0}^{n-1} \sup_{t\in I_i} B_t - B_{i/n} > \eta \right)$$

$$\leq \sum_{i=0}^{n-1} \mathbb{P}\left( \sup_{t\in I_i} B_t - B_{i/n} > \eta \right)$$

$$= \sum_{i=0}^{n-1} 2\mathbb{P}(B_{(i+1)/n} - B_{i/n} > \eta)$$

$$= \sum_{i=0}^{n-1} 2\mathbb{P}(\mathcal{N}(0, \sigma^2/n) > \eta)$$

$$\leq \sum_{i=0}^{n-1} \frac{2\exp\left(-\frac{\eta^2 n}{2\sigma^2}\right)}{\eta/\sigma\sqrt{2n\pi}} \leq \frac{2\sqrt{n}\exp\left(-\frac{\eta^2 n}{2\sigma^2}\right)}{\eta/\sigma\sqrt{2\pi}},$$

where the second equality holds from the reflection principle (see [Baldi, 2017]), and last inequality holds since it is well known that:

$$\mathbb{P}(\mathcal{N}(0, \beta^2) > y) \leq \frac{\exp(-y^2/2\beta^2)}{y/\beta\sqrt{2\pi}}$$

for tail bound on Gaussian distributions. In the exact same way, we can prove that:

$$\mathbb{P}\left(\inf_{i \in \{0,\ldots,n-1\}} (\inf_{t \in I_i} B_t - B_{i/n}) \leq -\eta\right) \leq \frac{2\sqrt{n}\exp\left(-\frac{\eta^2 n}{2\sigma^2}\right)}{\eta/\sigma\sqrt{2\pi}}.$$

Together, the two results imply the thesis. $\qquad\square$

We are now ready to prove a theorem that links Brownian motion and random walks in terms of the probability that each of them stays in the interval $[0, \infty)$.

**Lemma 5** (Discretization lemma). *Let $\{G_i\}_{i \in \{0,\ldots n-1\}}$ be a Gaussian $0$-mean unit variance random walk, $\mu \in \mathbb{R}$, and $\{B_t\}_{t \in [0,1]}$ a standard Brownian motion. Then, for every $s \in (0, 1)$, we have:*

$$\mathbb{P}\left(|B_t + t\mu_0 > \eta| > s\right) - P(n, \eta) \leq \mathbb{P}\left(\sum_{i=0}^{n-1} \mathbf{1}_{(0,\infty)}(G_i + i\mu) > sn\right)$$

$$\leq \mathbb{P}\left(|B_t + t\mu_0 > -\eta| > s\right) + P(n, \eta),$$

*and,*

$$\mathbb{P}\left(|B_t + t\mu_0 \leq \eta| \leq s\right) - P(n, \eta) \leq \mathbb{P}\left(\sum_{i=0}^{n-1} \mathbf{1}_{(-\infty,0]}(G_i + i\mu) \leq sn\right)$$

$$\leq \mathbb{P}\left(|B_t + t\mu_0 \leq -\eta| \leq s\right) + P(n, \eta),$$

*with $P(n, \eta) = \frac{2\sqrt{n}\exp(-\eta^2 n/2)}{\eta\sqrt{2\pi}}$ and $\mu_0 = \sqrt{n}\mu$.*

*Proof.* We only prove the first part, as the second one follows trivially by substituting:

$$s \leftarrow 1 - s, \; \mu \leftarrow -\mu, \; G_i \leftarrow -G_i, \; B_t \leftarrow -B_t.$$

Let $\{B_t\}_{t \in [0,1]}$ be a standard Brownian motion. We define:

$$I_i := [i/n, (i+1)/n],$$

for all $i \in \{0, \ldots, n-1\}$. Furthermore, we set $\mu_0 = \sqrt{n}\mu$. With this definition, we have the following set of inclusions for any $s \in [0, 1]$ and $\eta > 0$:

$$\{|B_t + t\mu_0 > \eta| > s\} = \left\{\int_0^1 \mathbf{1}_{(\eta,\infty)}(B_\tau + \tau\mu_0)\, d\tau > s\right\}$$

$$= \left\{\sum_{i=0}^{n-1} \int_{I_i} \mathbf{1}_{(\eta,\infty)}(B_\tau + \tau\mu_0)\, d\tau > s\right\}$$

$$\subseteq \left\{\sum_{i=0}^{n-1} \sup_{\tau \in I_i} \mathbf{1}_{(\eta,\infty)}(B_\tau + \tau\mu_0) > sn\right\}$$

$$\subseteq \left\{\sum_{i=0}^{n-1} \mathbf{1}_{(0,\infty)}\left(B_{i/n} + \frac{i}{n}\mu_0\right) > sn\right\} \cup \left\{\sup_{i \in \{0,\ldots,n-1\}}\left(\sup_{t \in I_i} B_t - B_{i/n}\right) \geq \eta\right\}.$$

Moreover, using the same steps above, it also holds:

$$\{|B_t + t\mu_0 > -\eta| > s\} = \left\{\int_0^1 \mathbf{1}_{(-\eta,\infty)}(B_\tau + \tau\mu_0)\, d\tau > s\right\}$$

$$\supseteq \left\{\sum_{i=0}^{n-1} \mathbf{1}_{(0,\infty)}\left(B_{i/n} + \frac{i}{n}\mu_0\right) > sn\right\} \cap \left\{\inf_{i \in \{0,\ldots,n-1\}}\left(\inf_{t \in I_i} B_t - B_{i/n}\right) \geq -\eta\right\}.$$

Now, note that the random variable $B_{i/n}$ for each $i \in [n]$ has the same distribution of $G_i/\sqrt{n}$, thus:

$$\mathbb{P}\left(\sum_{i=0}^{n-1} \mathbf{1}_{(0,\infty)}\left(B_{i/n} + \frac{i}{n}\mu_0\right) > sn\right) = \mathbb{P}\left(\sum_{i=0}^{n-1} \mathbf{1}_{(0,\infty)}\left(\sqrt{n}B_{i/n} + \frac{i}{\sqrt{n}}\mu_0\right) > sn\right)$$

$$= \mathbb{P}\left(\sum_{i=0}^{n-1} \mathbf{1}_{(0,\infty)}\left(G_i + i\mu\right) > sn\right).$$

Therefore, by union bound, we have:

$$\mathbb{P}\left(|B_t + t\mu_0 > \eta| > s\right) \le \mathbb{P}\left(\sum_{i=0}^{n-1} \mathbf{1}_{(0,\infty)}\left(G_i + i\mu\right) > sn\right) + \mathbb{P}\left(\sup_{i \in \{0,\ldots,n-1\}}\left(\sup_{t \in I_i} B_t - B_{i/n}\right) \ge \eta\right),$$

and,

$$\mathbb{P}\left(|B_t + t\mu_0 > -\eta| > s\right) \ge \mathbb{P}\left(\sum_{i=0}^{n-1} \mathbf{1}_{(0,\infty)}\left(G_i + i\mu\right) > sn\right) - \mathbb{P}\left(\inf_{i \in \{0,\ldots,n-1\}}\left(\inf_{t \in I_i} B_t - B_{i/n}\right) \le -\eta\right).$$

The proof is completed applying Lemma 4 and reordering the terms. $\qquad\square$

**Corollary 5.** *Let $\{G_i\}_{i \in \{0,\ldots n-1\}}$ be a Gaussian $0$-mean unit variance random walk, and $\mu \in \mathbb{R}$. Then, for every $s \in (0,1)$, we have:*

$$\mathbb{P}\left(\sum_{i=0}^{n-1} \mathbf{1}_{(-\infty,0]}\left(G_i + i\mu\right) \le sn\right) \in \left[\mathbb{P}\left(\left|B_t + t\mu_0 \le \frac{2\log(n)}{\sqrt{n}}\right| \le s\right) - \frac{\sqrt{2}}{\sqrt{\pi}n\log(n)},\right.$$

$$\left.\mathbb{P}\left(\left|B_t + t\mu_0 \le -\frac{2\log(n)}{\sqrt{n}}\right| \le s\right) + \frac{\sqrt{2}}{\sqrt{\pi}n\log(n)}\right],$$

*where $\mu_0 = \sqrt{n}\mu$.*

*Proof.* It is sufficient to make the substitution:

$$\eta = \frac{2\log(n)}{\sqrt{n}},$$

in Lemma 5. Then, we have:

$$P(n, \eta) = \frac{2\sqrt{n}\exp\left(-\eta^2 n/2\right)}{\eta\sqrt{2\pi}}$$

$$= \frac{2\sqrt{n}\exp\left(-\log(n)^2\right)}{\frac{\log(n)}{\sqrt{n}}\sqrt{2\pi}}$$

$$= \frac{2n\exp\left(-\log(n)^2\right)}{\log(n)\sqrt{2\pi}} = \frac{\sqrt{2}}{\sqrt{\pi}n\log(n)},$$

concluding the proof. $\qquad\square$

### B.2.2 Proofs of filtering inequalities

All the proof of this subsections will be based on the following very powerful result, which studies the time spent by a Brownian Motion with drift in the half-line $[0,\infty)$.

**Theorem 6** (Takács [1996]). *Let $B_t$ be a standard Brownian motion on $t \in [0,1]$, and let us note as $|\cdot|$ the Lebesgue measure of a set. For $\mu_0 \in \mathbb{R}$ and $\eta > 0$, we have*

$$\mathbb{P}\left(|B_t + t\mu_0 \le \eta| \le s\right) = 2\int_0^s \left[\frac{\varphi(\mu_0\sqrt{1-\tau})}{\sqrt{1-\tau}} + \mu_0\Phi(\mu_0\sqrt{1-\tau})\right] \times$$

$$\left[ \frac{\varphi(\eta/\sqrt{\tau} - \mu_0\sqrt{\tau})}{\sqrt{\tau}} - \mu_0 e^{2\mu_0\eta}\Phi(-\eta/\sqrt{\tau} - \mu_0\sqrt{\tau}) \right] d\tau,$$

*where*

$$\varphi(x) := \frac{1}{\sqrt{2\pi}}e^{-x^2/2} \qquad \Phi(x) := \int_{-\infty}^{x} \varphi(u) \, du.$$

Thanks to the previous theorem, we can prove the following crucial results.

**Theorem 7.** *Let $T$ be a sufficiently large constant. Let $\{G_i\}_{i \in \{0,\ldots n-1\}}$ be a Gaussian $0$-mean unit variance random walk, and $\mu \in \mathbb{R}$. If $\mu \geq CT^{-\alpha}$, for some $\alpha \in (0, 1/2)$ and $C = 4\log(T)$, then setting $n = \lceil T^{1/2+\alpha} \rceil$ we have*

$$\mathbb{P}\left( \sum_{i=0}^{n-1} \mathbf{1}_{(-\infty,0]} (G_i + i\mu) \leq T^{2\alpha} \right) \geq 1 - 2T^{-1/2}.$$

*Proof.* In the rest of the proof, we will assume, for ease of notation, that $T$ is such that $T^{1/2+\alpha}$ an integer, so that $n = T^{1/2+\alpha}$. This is done without loss of generality, since substituting $n$ with $n+1$ leads to a negligble difference for $T$ sufficiently big. Applying the discretization corollary 5, we have that for every $s \in (0, 1)$

$$\mathbb{P}\left( \sum_{i=0}^{n-1} \mathbf{1}_{(-\infty,0]} (G_i + i\mu) \leq sn \right) \geq \mathbb{P}\left( \left| B_t + t\mu_0 \leq \frac{2\log(n)}{\sqrt{n}} \right| \leq s \right) - \frac{\sqrt{2}}{\sqrt{\pi}n\log(n)}, \quad (6)$$

where $\mu_0 = \sqrt{n}\mu$. Therefore, by assumption,

$$\mu_0 = \sqrt{n}\mu \geq (T^{1/2+\alpha})^{1/2}CT^{-\alpha} = CT^{1/4-\alpha/2}.$$

At this point, we can apply Theorem 6 to have, for any $\eta > 0$,

$$\mathbb{P}\left( |B_t + t\mu_0 \leq \eta| \leq s \right) = 2\int_0^s \left( \frac{\phi(\mu_0\sqrt{1-\tau})}{\sqrt{1-\tau}} + \mu_0\Phi(\mu_0\sqrt{1-\tau}) \right)$$
$$\times \left( \phi\left( \frac{\eta - \mu_0\tau}{\sqrt{\tau}} \right) \frac{1}{\sqrt{\tau}} - \mu_0 e^{2\mu_0\eta}\Phi\left( \frac{-\eta - \mu_0\tau}{\sqrt{\tau}} \right) \right) d\tau$$

which means that

$$\mathbb{P}\left( |B_t + t\mu_0 \leq \eta| \leq s \right) = 1 - 2\int_s^1 \left( \underbrace{\frac{\phi(\mu_0\sqrt{1-\tau})}{\sqrt{1-\tau}}}_{(1)} + \underbrace{\mu_0\Phi(\mu_0\sqrt{1-\tau})}_{(2)} \right)$$
$$\times \left( \underbrace{\phi\left( \frac{\eta - \mu_0\tau}{\sqrt{\tau}} \right) \frac{1}{\sqrt{\tau}}}_{(3)} - \underbrace{\mu_0 e^{2\mu_0\eta}\Phi\left( \frac{-\eta - \mu_0\tau}{\sqrt{\tau}} \right)}_{(4)} \right) d\tau.$$

Here, we have to consider that

- $\eta = \frac{2\log(n)}{\sqrt{n}} \leq 2\log(T)T^{-\alpha/2-1/4}$

- $\mu_0 \geq CT^{1/4-\alpha/2}$.

Moreover, to have the thesis, we are interested in a value of $s$ such that $sn = T^{2\alpha}$, corresponding to $T^{-1/2+\alpha}$. Therefore, in the interval $[T^{-1/2+\alpha}, 1]$, we have

1. Consider term (3):

$$\phi\left(\frac{\eta - \mu_0\tau}{\sqrt{\tau}}\right)\frac{1}{\sqrt{\tau}} \leq \phi\left(\frac{\eta - \mu_0 T^{-1/2+\alpha}}{T^{-1/4+\alpha/2}}\right)\frac{1}{T^{-1/4+\alpha/2}}$$

$$= \phi\left(\eta T^{1/4-\alpha/2} - \mu_0 T^{-1/4+\alpha/2}\right)\frac{1}{T^{-1/4+\alpha/2}}.$$

Here, since $\eta = \frac{2\log(n)}{\sqrt{n}} \leq 2\log(T)T^{-\alpha/2-1/4}$, the part $\eta T^{1/4-\alpha/2}$ is bounded by $2\log(T)$.

Instead, $\mu_0 T^{-1/4+\alpha/2} \geq CT^{1/4-\alpha/2}T^{-1/4+\alpha/2} = C$.

2. Term (4) is non-negative.

Therefore, for $C = 4\log(T)$, we have that in the interval $[T^{-1/2+\alpha}, 1]$

$$(3) + (4) \leq \phi\left(2\log(T)\right)\frac{1}{T^{-1/4+\alpha/2}} = \frac{1}{\sqrt{2\pi}T^{-1/4+\alpha/2}}e^{-2\log(T)^2} \leq \frac{T^{-1}}{\sqrt{2\pi}}.$$

With this inequality, we have

$$\mathbb{P}\left(|B_t + t\mu_0 \leq \eta| \leq T^{-1/2+\alpha}\right) = 1 - 2\frac{T^{-1}}{\sqrt{2\pi}}\int_{T^{-1/2+\alpha}}^{1}\left(\frac{\phi(\mu_0\sqrt{1-\tau})}{\sqrt{1-\tau}} + \mu_0\Phi(\mu_0\sqrt{1-\tau})\right)d\tau$$

$$\geq 1 - 2\frac{T^{-1}}{\sqrt{2\pi}}\int_{T^{-1/2+\alpha}}^{1}\frac{1}{\sqrt{2\pi(1-\tau)}} + |\mu_0|d\tau$$

$$\geq 1 - 2\frac{T^{-1}}{\sqrt{2\pi}}\int_{0}^{1}\frac{1}{\sqrt{2\pi(1-\tau)}} + |\mu_0|d\tau$$

$$= 1 - 2\frac{T^{-1}}{\sqrt{2\pi}}\left(\frac{\sqrt{2}}{\sqrt{\pi}} + \mu_0\right).$$

At this point, knownig from the assumptions that $n < T$, we have $\mu_0 \leq \sqrt{T}$, which implies

$$\mathbb{P}\left(|B_t + t\mu_0 \leq \eta| \leq T^{-1/2+\alpha}\right) \geq 1 - \frac{T^{-1/2}}{\pi}.$$

Substituting this result into Equation 6, we get, for $s = T^{-1/2+\alpha}$ and $n \geq T^{1/2+\alpha}$

$$\mathbb{P}\left(\sum_{i=0}^{n-1}\mathbf{1}_{(-\infty,0]}\left(G_i + i\mu\right) < T^{2\alpha}\right) \geq 1 - \frac{T^{-1/2}}{\pi} - \frac{\sqrt{2}}{\sqrt{\pi}T^{1/2+\alpha}\log(T^{1/2+\alpha})}$$

$$\geq 1 - 2T^{-1/2}.$$

$\square$

The second result is the following

**Theorem 8.** *Let $T$ be a sufficiently large constant. Let $\{G_i\}_{i\in\{0,...n-1\}}$ be a Gaussian 0-mean unit variance random walk, and $\mu \in \mathbb{R}$ such that $\mu \leq -CT^{-\theta}$, for some $\theta \in (0, 1/2)$ and $C = 2\sqrt{\log(T)} + 2$. Then, for any $\alpha \in (0, 1/2)$, setting $n = \lfloor T^{1/2+\alpha} \rfloor$ we have*

$$\mathbb{P}\left(\sum_{i=0}^{n-1}\mathbf{1}_{(-\infty,0]}\left(G_i + i\mu\right) \leq T^{2\alpha}\right) \leq 3T^{-1/2+\theta}.$$

*Proof.* In the rest of the proof, we will assume, for ease of notation, that $T$ is such that $T^{1/2+\alpha}$ an integer, so that $n = T^{1/2+\alpha}$. This is done without loss of generality, since substituting $n$ with $n+1$ leads to a negligble difference for $T$ sufficiently large. Applying the discretization corollary 5, we have that for every $s \in (0,1)$

$$\mathbb{P}\left(\sum_{i=0}^{n-1} \mathbf{1}_{(-\infty,0]}(G_i + i\mu) \leq sn\right) \leq \mathbb{P}\left(\left|B_t + t\mu_0 \leq -\frac{2\log(n)}{\sqrt{n}}\right| \leq s\right) + \frac{\sqrt{2}}{\sqrt{\pi}n\log(n)}, \quad (7)$$

where $\mu_0 = \sqrt{n}\mu$. Therefore, by assumption,

$$\mu_0 = \sqrt{n}\mu \leq -(T^{1/2+\alpha})^{1/2}CT^{-\theta} = -CT^{1/4+\alpha/2-\theta}.$$

Differently from the previous proof, here we cannot directly apply Theorem 6, since $\eta = -\frac{2\log(n)}{\sqrt{n}} < 0$.

Still, we can say that

$$\mathbb{P}\left(\left|B_t + t\mu_0 \leq -\frac{2\log(n)}{\sqrt{n}}\right| \leq s\right) = \mathbb{P}\left(\left|-B_t - t\mu_0 > \frac{2\log(n)}{\sqrt{n}}\right| \leq s\right)$$
$$= \mathbb{P}\left(\left|-B_t - t\mu_0 \leq \frac{2\log(n)}{\sqrt{n}}\right| > 1 - s\right).$$

At this point, we set $\eta = \frac{2\log(n)}{\sqrt{n}}$, $\tilde{\mu}_0 = -\mu_0$ and $B_t = -B_t$ (it is not necessary to rename it since its distribution is symmentric). In this way we can apply Theorem 6 having that the previous probability corresponds to

$$\mathbb{P}(|B_t + t\tilde{\mu}_0 \leq \eta| > 1 - s) = 2\int_{1-s}^{1}\left(\underbrace{\frac{\phi(\tilde{\mu}_0\sqrt{1-\tau})}{\sqrt{1-\tau}}}_{(1)} + \underbrace{\tilde{\mu}_0\Phi(\tilde{\mu}_0\sqrt{1-\tau})}_{(2)}\right)$$

$$\times\left(\underbrace{\phi\left(\frac{\eta - \tilde{\mu}_0\tau}{\sqrt{\tau}}\right)\frac{1}{\sqrt{\tau}}}_{(3)} - \underbrace{\tilde{\mu}_0 e^{2\mu_0\eta}\Phi\left(\frac{-\eta - \tilde{\mu}_0\tau}{\sqrt{\tau}}\right)}_{(4)}\right) d\tau.$$

Here, we have to consider that

- $\eta = \frac{2\log(n)}{\sqrt{n}} \leq 2\log(T)T^{-\alpha/2-1/4}$

- $\tilde{\mu}_0 \geq CT^{1/4+\alpha/2-\theta}$.

Moreover, to have the thesis, we are interested in a value of $s$ such that $sn = T^{2\alpha}$, corresponding to $T^{-1/2+\alpha}$.

Here, it is convenient to divide the proof in two cases, depending on the sign of $1/4 + \alpha/2 - \theta$.

1. Assume $(1/4 + \alpha/2 - \theta > 0)$. Then, considering term (3) we have that for $\tau \in [1/2, 1]$

$$(3) \leq \phi\left(\frac{\eta - \tilde{\mu}_0\tau}{\sqrt{\tau}}\right)\frac{1}{\sqrt{\tau}} \leq \sqrt{2}\phi\left(\sqrt{2}\eta - \tilde{\mu}_0/\sqrt{2}\right).$$

Moreover, since term (4) is nonnegative we also have

$$(3) + (4) \leq \sqrt{2}\phi\left(\sqrt{2}\eta - \tilde{\mu}_0/\sqrt{2}\right) = \frac{1}{\sqrt{\pi}}e^{-(\sqrt{2}\eta-\tilde{\mu}_0/\sqrt{2})^2/2}.$$

Being $1/4 + \alpha/2 - \theta > 0$ and $\eta < 1$, the exponent is less than $-(\sqrt{2} - C/\sqrt{2})^2/2$. This means that for $C = 2\sqrt{\log(T)} + 2$ the full term is bounded by

$$(3) + (4) \leq \frac{1}{\sqrt{\pi}} e^{-(\sqrt{2} - C/\sqrt{2})^2/2} = \frac{1}{\sqrt{\pi}} e^{-(\sqrt{2\log(T)})^2/2} = \frac{T^{-1}}{\sqrt{\pi}}.$$

Substituting this inequality, we get

$$\mathbb{P}\left(|B_t + t\tilde{\mu}_0 \leq \eta| > 1 - T^{-1/2+\alpha}\right) \leq \frac{2T^{-1}}{\sqrt{\pi}} \int_{1-T^{-1/2+\alpha}}^{1} \left(\frac{\phi(\tilde{\mu}_0\sqrt{1-\tau})}{\sqrt{1-\tau}} + \tilde{\mu}_0 \Phi(\tilde{\mu}_0\sqrt{1-\tau})\right) d\tau$$

$$\leq \frac{2T^{-1}}{\sqrt{\pi}} \int_{1-T^{-1/2+\alpha}}^{1} \frac{1}{\sqrt{2\pi(1-\tau)}} + |\tilde{\mu}_0| d\tau$$

$$\leq \frac{2T^{-1}}{\sqrt{\pi}} (2 + T^{-1/2+\alpha}\tilde{\mu}_0) \leq \frac{6T^{-1}}{\sqrt{\pi}}.$$

This quantity is of course less than $T^{-\theta}$, since $\theta \in (0, 1/2)$ by assumption

2. Assume $(1/4 + \alpha/2 - \theta < 0)$. In this case, we have, being $\tilde{\mu}_0 \geq 0$, the following inequality

$$\mathbb{P}\left(|B_t + t\tilde{\mu}_0 \leq \eta| > 1 - s\right) \leq \mathbb{P}\left(|B_t \leq \eta| > 1 - s\right).$$

This simplified form leads to

$$\mathbb{P}\left(|B_t + t\tilde{\mu}_0 \leq \eta| > 1 - s\right) \leq 2 \int_{1-s}^{1} \frac{\phi(0)}{\sqrt{1-\tau}} \phi\left(\frac{\eta}{\sqrt{\tau}}\right) \frac{1}{\sqrt{\tau}} d\tau$$

$$\leq 2 \int_{1-s}^{1} \frac{\phi(0)}{\sqrt{1-\tau}} \phi(0) \frac{1}{\sqrt{\tau}} d\tau$$

$$= \frac{1}{\pi} \int_{1-s}^{1} \frac{1}{\sqrt{\tau(1-\tau)}} d\tau.$$

Since in our case $s = T^{-1/2+\alpha} < 1/2$, this can be further simplified as

$$\mathbb{P}\left(|B_t + t\tilde{\mu}_0 \leq \eta| > 1 - s\right) \leq \frac{1}{\pi} \int_{1-s}^{1} \frac{1}{\sqrt{\tau(1-\tau)}} d\tau$$

$$= \frac{2}{\pi} \int_{1-s}^{1} \frac{1}{\sqrt{1-\tau}} d\tau$$

$$\stackrel{y=1-\tau}{=} \frac{2}{\pi} \int_{0}^{s} \frac{1}{\sqrt{y}} dy = \frac{4}{\pi} \sqrt{s}.$$

This leads to

$$\mathbb{P}\left(|B_t + t\tilde{\mu}_0 \leq \eta| > 1 - T^{-1/2+\alpha}\right) \leq \frac{4}{\pi} T^{-1/4+\alpha/2}.$$

By assumption, $1/4 + \alpha/2 - \theta < 0$ the exponent is $-1/4 + \alpha/2 < T^{-1/2+\theta}$. Therefore, we have

$$\mathbb{P}\left(|B_t + t\tilde{\mu}_0 \leq \eta| > 1 - T^{-1/2+\alpha}\right) \leq \frac{4}{\pi} T^{-1/2+\theta}.$$

Thus, we have proved that in both cases

$$\mathbb{P}\left(|B_t + t\tilde{\mu}_0 \leq \eta| > 1 - s\right) \leq \frac{4}{\pi} T^{-1/2+\theta}.$$

Finally, applying Equation (7) and substituting the value of $n$, we get

$$\mathbb{P}\left(\sum_{i=0}^{n-1}\mathbf{1}_{(-\infty,0]}\left(G_i + i\mu\right) \leq T^{2\alpha}\right) \leq \frac{4}{\pi}T^{-1/2+\theta} + \frac{\sqrt{2}}{\sqrt{\pi}T^{1/2+\alpha}\log(T^{1/2+\alpha})},$$

which implies

$$\mathbb{P}\left(\sum_{i=0}^{n-1}\mathbf{1}_{(-\infty,0]}\left(G_i + i\mu\right) \leq T^{2\alpha}\right) \leq 3T^{-1/2+\theta}.$$

$\square$

### B.2.3 Regret bound

Before the actual proof, we are stating a simple proposition about the structure of the loggrid, which will ease the next computations.

**Proposition 9.** *Let*

$$LG(1/2, 1, T) := \left\{ \lfloor T^{\lambda_j + (1-\lambda_j)/2} \rfloor : \lambda_j = \frac{j}{\lfloor \log(T) \rfloor}, \; \forall j = 0, \ldots, \lfloor \log(T) \rfloor \right\}.$$

*The following identities hold*

1. *$LG(1/2, 1, T)$ can be equivalently defined as*

   $$LG(1/2, 1, T) := \left\{ \lfloor T^{1/2 + \frac{j}{2\lfloor \log(T) \rfloor}} \rfloor, \; \forall j = 0, \ldots, \lfloor \log(T) \rfloor \right\}.$$

2. *Let $\ell_j$ the $j-$th element of $LG(1/2, 1, T)$, and $\alpha_j = \frac{\log(\ell_j)}{\log(T)} - 1/2$. Then $\alpha_j = \frac{j}{2\lfloor \log(T) \rfloor} + o(T^{-1/2})$.*

3. *The ratio of two consecutive values of $\ell_j$ is $\frac{\ell_{j+1}}{\ell_j} \approx T^{\frac{1}{2\lfloor \log(T) \rfloor}} \in [\sqrt{e}, 2]$ for $T \geq 51$.*

Next, we prove the following lemmas, which concern some features of our algorithm.

**Lemma 6.** *For any arm $i$, the probability of the event $E_{ii_0}$, corresponding to $i$ eliminating the another $i_0$ arm such that their gap is $\Delta_{ii_0} := \mu_{i_0} - \mu_i > 0$, is, at most*

$$\mathbb{P}(E_{ii_0}) \leq 6\log(T)(4\log(T) + 2)T^{-1/2}\Delta_{ii_0}^{-1}.$$

*Proof.* Let us call:

$$\widetilde{\Delta}_{ii_0} = \frac{\Delta_{ii_0}}{4\log(T) + 2}.$$

At this point, there are two possibilities,

1. $\widetilde{\Delta}_{ii_0} \leq T^{-1/2}$: in this case, the statement of the lemma is vacuous.

2. $\widetilde{\Delta}_{ii_0} > T^{-1/2}$: in this case, by assumption, there are two consecutive $\ell_{j_\star}, \ell_{j_\star+1} \in \mathcal{L}$ such that

   $$\widetilde{\Delta}_{ii_0} \in \left( \frac{T^{1/2}}{\ell_{j_\star+1}}, \frac{T^{1/2}}{\ell_{j_\star}} \right].$$

   This is true due to the fact that that the sequence $\ell_j$ spans from $T^{1/2}$ to $T$. By Proposition 9, this can be equivalently expressed by saying that

   $$\widetilde{\Delta}_{ii_0} \in \left( T^{-\frac{j_\star+1}{2\lfloor \log(T) \rfloor}}, T^{-\frac{j_\star}{2\lfloor \log(T) \rfloor}} \right].$$

Let us define the following family of events:

$$E_{ii_0}(z) := \text{arm } i \text{ eliminates arm } i_0 \text{ when both have been pulled } z \text{ times.}$$

The probability of $E_{ii_0}$ is bounded by:

$$\mathbb{P}(E_{ii_0}) = \mathbb{P}\left(\bigcup_{z=1}^{T} E_{ii_0}(z)\right) = \mathbb{P}\left(\bigcup_{z \in \mathcal{L}} E_{ii_0}(z)\right)$$

$$\leq \sum_{j=1}^{|\mathcal{L}|} \mathbb{P}(E_{ii_0}(\ell_j)).$$

Here, we have applied the fact that, by design of the algorithm, the arms can only be discarded in fair steps for which $z \in \mathcal{L}$ and then a union bound. Here, we notice that by definition of the filtering condition, defining $t_j$ as the fair time-step where both arms have been played $\ell_j$ times, this event can be again rewritten as:

$$\left\{\sum_{\tau=1:\tau \text{ fair}}^{t_j} \{\mathcal{R}_\tau(i) > \mathcal{R}_\tau(i_0)\} \geq T^{2\alpha_j}\right\},$$

where $\alpha_j = \frac{\log(\ell_j)}{\log(T)} - \frac{1}{2}$. If we call $\hat{\mu}_{\tau,i_0}, \hat{\mu}_{\tau,i}$ the empirical means of arms $i_0, i$ after $\tau$ pulls of each, the previous event can be interpreted as the time in which the random walk given by the difference of the rewards of the two arms stays in $(-\infty, 0]$:

$$\sum_{\tau=1:\tau \text{ fair}}^{t_j} \{\mathcal{R}_\tau(i) > \mathcal{R}_\tau(i_0)\} = \sum_{\tau=1}^{\ell_j} \mathbf{1}\left\{\hat{\mu}_{\tau,i} \geq \hat{\mu}_{\tau,i_0}\right\}$$

$$= \sum_{\tau=1}^{\ell_j} \mathbf{1}_{(-\infty,0]}\left(\underbrace{\sum_{k=1}^{\tau} r_{i,k} - \sum_{j=1}^{\tau} r_{1,k}}_{G_\tau}\right),$$

where $\sum_{k=1}^{\tau} r_{i,k}$ is the cumulative reward of arm $i$ and $\sum_{k=1}^{\tau} r_{1,k}$ is the cumulative reward of arm $i_0$. Therefore, we have written this quantity as the time spent by the random walk $G_\tau$ in the interval $(-\infty, 0]$, for $\tau = 1, \ldots \ell_j$. The drift term for this random walk is given by:

$$\mathbb{E}[r_{i,k} - r_{1,k}] = \mu_{i_0} - \mu_i = -\Delta_{ii_0}.$$

Therefore, we can apply Theorem 8 for the following choice of parameters,

(a) $\alpha = \alpha_j = \frac{j}{2\lfloor \log(T) \rfloor} + o(T^{-1/2})$ (Proposition 9), which implies $n = \lfloor T^{1/2+\alpha_j} \rfloor = \ell_j$.

(b) $\theta = \frac{j_\star+1}{2\lfloor \log(T) \rfloor}$. We can use this choice since the drift is

$$-\Delta_{ii_0} = -\underbrace{(4\log(T) + 2)}_{\geq 2\sqrt{\log(T)}+2}\underbrace{\widetilde{\Delta}_{ii_0}}_{\geq T^{-\frac{j_\star+1}{2\lfloor \log(T) \rfloor}}},$$

therefore the assumptions of the theorem are respected.

Applying the theorem, we have:

$$\mathbb{P}(E_i(2\ell_j)) \leq 3T^{-1/2+\theta} = 3T^{-1/2+\frac{j_\star+1}{2\lfloor \log(T) \rfloor}}.$$

Summing over $j$, we get,

$$\mathbb{P}\left(\bigcup_{t=1}^{T} E_1(t)\right) \leq \sum_{j=1}^{|\mathcal{L}|} \mathbb{P}(E_1(2\ell_j))$$

$$\leq 3\log(T)T^{-1/2+\frac{j_\star+1}{2\lfloor \log(T) \rfloor}}.$$

To conclude, consider that, by definition ,

$$\ell_j = T^{1/2 + \frac{j}{2\lfloor \log(T)\rfloor}};$$

this means that:

$$\widetilde{\Delta}_{ii_0} \in \left( T^{-\frac{j_\star+1}{2\lfloor \log(T)\rfloor}}, T^{-\frac{j_\star}{2\lfloor \log(T)\rfloor}} \right],$$

so that, in particular,

$$\widetilde{\Delta}_{ii_0}^{-1} \geq T^{\frac{j_\star}{2\lfloor \log(T)\rfloor}}$$

and

$$\Delta_{ii_0}^{-1} \geq (4\log(T) + 2) T^{\frac{j_\star}{2\lfloor \log(T)\rfloor}}.$$

Substituting in the bound we have just found results in,

$$3\log(T) T^{-1/2 + \frac{j_\star+1}{2\lfloor \log(T)\rfloor}} \leq 3\log(T)(4\log(T) + 2) T^{-1/2} T^{\frac{1}{2\lfloor \log(T)\rfloor}} \Delta_{ii_0}^{-1}$$

$$\leq 6\log(T)(4\log(T) + 2) T^{-1/2} \Delta_{ii_0}^{-1}.$$

$\square$

**Lemma 7.** *For any arm $i \in [n]$, the probability of the event $E_{ii_0}^*$, corresponding to arm $i \in [n]$ not being eliminated after $(8\log(T) + 4) T^{1/2} \Delta_{ii_0}^{-1}$ pulls if there is an active arm $i_0$ with $\Delta_{ii_0} := \mu_{i_0} - \mu_i > 0$ is such that:*

$$\mathbb{P}(E_{ii_0}^*) \leq 2T^{-1/2}.$$

*Proof.* Define $\ell_j, \alpha_j$ as in the previous lemma, so that $\alpha_j = \frac{\log(\ell_j)}{\log(T)} - \frac{1}{2}$. As in the previous lemma, we define:

$$\widetilde{\Delta}_{ii_0} = \frac{\Delta_{ii_0}}{4\log(T) + 2},$$

and $j_\star \in \{1, \dots \lfloor \log(T)\rfloor\}$ such that:

$$\widetilde{\Delta}_{ii_0} \in \left( T^{-\frac{j_\star+1}{2\lfloor \log(T)\rfloor}}, T^{-\frac{j_\star}{2\lfloor \log(T)\rfloor}} \right].$$

Here, remember that by definition of the filtering condition, defining $t_{j_\star}$ as the fair timestep where both arms have been played $\ell_{j_\star}$ times, $E_{i^*i_0}$ is included in the following event:

$$\left\{ \sum_{\tau=1:\tau \text{ fair}}^{t_{j_\star}} \{\mathcal{R}_\tau(i_0) > \mathcal{R}_\tau(i)\} \geq T^{2\alpha_{j_\star}} \right\}.$$

As before, this event can be interpreted as the difference between two random walks being negative, due to the fact that:

$$\sum_{\tau=1:\tau \text{ fair}}^{t_{j_\star}} \{\mathcal{R}_\tau(i_0) > \mathcal{R}_\tau(i)\} = \sum_{\tau=1}^{\ell_{j_\star}} \mathbf{1}\{\hat{\mu}_{\tau,1} \geq \hat{\mu}_{\tau,i}\}$$

$$= \sum_{\tau=1}^{\ell_{j_\star}} \mathbf{1}_{(-\infty,0]} \left( \underbrace{\sum_{k=1}^{\tau} r_{i_0,k} - \sum_{j=1}^{\tau} r_{i,k}}_{G_\tau} \right).$$

In this formulation, we have written the quantity of interest for the filtering condition after $\ell_{j_\star+1}$ pulls as the time spent by the random walk $G_\tau$ in the interval $(-\infty, 0]$, for $\tau = 1, \dots \ell_{j_\star+1}$. This time, the drift term is:

$$\mathbb{E}[r_{i_0,k} - r_{i,k}] = \mu_{i_0} - \mu_i = \Delta_{ii_0}.$$

Therefore, we can apply Theorem 7 for $\alpha = \alpha_{j_\star}$, since, by assumption,

$$\Delta_{ii_0} = \underbrace{(4\log(T) + 2)}_{\geq 4\log(T)} \underbrace{\widetilde{\Delta}_{ii_0}}_{\geq T^{-\frac{j_\star+1}{2\lfloor \log(T)\rfloor}}}.$$

This theorem leads to:

$$\mathbb{P}\left(E_{ii_0}^*\right) \leq 1 - \mathbb{P}\left(\sum_{\tau=1}^{\ell_{j_\star+1}} \mathbf{1}_{(-\infty,0]}\left(G_\tau\right) \leq T^{2\alpha_{j_\star+1}}\right)$$

$$\overset{\text{Thm.7}}{\leq} 2T^{-1/2}. \tag{8}$$

To get the thesis, is sufficient to reformulate the critical time-step $\ell_{j_\star+1}$ in terms of $\Delta_{ii_0}$. By definition,

$$\widetilde{\Delta}_{ii_0} \leq T^{-\frac{j_\star}{2\lfloor\log(T)\rfloor}} = \frac{T^{1/2}}{\ell_{j_\star}} \leq 2\frac{T^{1/2}}{\ell_{j_\star+1}}.$$

Therefore, $\Delta_{ii_0} \leq (8\log(T)+4)\frac{T^{1/2}}{\ell_{j_\star+1}}$. From this, it immediately follows,

$$\ell_{j_\star+1} \leq (8\log(T)+4)T^{1/2}\Delta_{ii_0}^{-1},$$

concluding the proof. $\qquad\square$

We are finally able to prove our main result about the instance-independent regret of the algorithm.

**Theorem 4.** *In the stochastic bandits with ranking feedback setting, when the noise is Gaussian, Algorithm 2 achieves* $R_T \leq 62n^4\log(T)^2T^{1/2}$.

*Proof.* Fix a sub-optimal arm $i$ with corresponding gap $\Delta_i$ with respect to the optimal arm and let $\psi$ a parameter to be chosen later. Define, for every couple of indices $i_1, i_0$ the event:

$$E_{i_1i_0}^\psi := \begin{cases} E_{i_1i_0} & \mu_{i_0} - \mu_{i_1} > \psi \\ \emptyset & \text{else,} \end{cases}$$

where the event $E_{i_1i_0}$ is defined as arm $i_1$ eliminating arm $i_0$ in some point of the process. The probability that at least one of this events verifies is bounded by lemma 6 and union bound with:

$$\mathbb{P}\left(\Psi\right) := \mathbb{P}\left(\bigcup_{i_1=1,i_0=0}^{n} E_{i_1i_0}^\psi\right) \leq 3n(n-1)\log(T)(4\log(T)+2)T^{-1/2}\psi^{-1},$$

as their number is at most $n(n-1)/2$. Therefore, under the complementary of $\Psi$, as no elimination with gap larger than $\psi$ happens, we are sure that an arm $i^\star$ with gap (w.r.t. the first arm) less than $(n-1)\psi$ survives until the last: in fact, at most $n-1$ eliminations may happen, and all between pair of arms with a difference at most $\psi$.

As the arm $i^\star$ is active until the last, the probability of event $E_{ii^\star}^*$ that $i$ survives for more than $(8\log(T)+4)T^{1/2}\Delta_{ii^\star}^{-1}$ pulls is bounded by Lemma 7 with $2T^{-1/2}$. Making the union bound over all possible values of $i^\star$, which is a random variable, we can say that the probability that any of this event happens is at most $2(n-1)T^{-1/2}$. Summarizing, we have the following bound on $Z_i(T)$:

1. $Z_i(T) \leq T$ if either $\Psi$ verifies, which happens with probability,

   $$3n(n-1)\log(T)(4\log(T)+2)T^{-1/2}\psi^{-1},$$

   or if $E_{ii^\star}$ verifies, which happens with probability at most $2(n-1)T^{-1/2}$

2. $Z_i(T) \leq T(8\log(T)+4)T^{1/2}(\Delta_i-(n-1)\psi)^{-1}$ otherwise. The last comes just from the fact that $\Delta_{ii^\star} \geq \Delta_i - (n-1)\psi$, by definition of $i^\star$.

If we take $\psi = \frac{\Delta}{2(n-1)}$, it results in,

$$\mathbb{E}[\Delta_i Z_i(T)] \leq \Delta_i T\mathbb{P}(\cup_{i_1=1,i_0=0}^n E_{i_1i_0}^\psi) + T\Delta_i\mathbb{P}(E_{ii^\star})$$
$$+ \Delta_i T(8\log(T)+4)T^{1/2}(\Delta_i-(n-1)\psi)^{-1}$$
$$\leq \Delta_i T\left(3n(n-1)\log(T)(4\log(T)+2)T^{-1/2}\psi^{-1} + 2nT^{-1/2}\right)$$

$$+ \Delta_i T(8\log(T) + 4)T^{1/2}(\Delta_i - (n-1)\psi)^{-1}$$

$$\overset{\psi = \frac{\Delta}{2(n-1)}}{\leq} \Delta_i T\left(6n(n-1)^2\log(T)(4\log(T)+2)T^{-1/2}\Delta_i^{-1} + 2nT^{-1/2}\right)$$

$$+ 2\Delta_i T(8\log(T) + 4)T^{1/2}\Delta_i^{-1}$$

$$= 6n(n-1)^2\log(T)(4\log(T)+2)T^{1/2} + 2n\Delta_i T^{1/2} + 2T(8\log(T)+4)T^{1/2}.$$

Being $\Delta_i \in (0,1)$ and $n \geq 2$, the previous quantity is bounded by,

$$\mathbb{E}[\Delta_i Z_i(T)] \leq 62n^3\log(T)^2 T^{1/2}.$$

The proof is completed by using the the Regret Decomposition Lemma Lattimore and Szepesvari [2017]. $\qquad\square$

## C  Proof for adversarial setting

**Theorem 5.** *In adversarial bandits with ranking feedback, there exists a constant $\gamma \in (0,1)$ such that no algorithm achieves $o(T)$ regret with respect to the best arm in hindsight with probability greater than $\gamma$.*

*Proof.* This negative result follows from the impossibility to achieve $R_T \leq CT$ regret by any algorithm, with $C$ properly set constant and probability $1 - \bar{\epsilon}$, in all three instances reported next. Please notice that, this result implies that even the no-regret property cannot be achieved in the bandit with ranking feedback setting.

Without loss of generality we consider rewards function bounded in $[0, 10]$. Consider three instances, with two arms $a_0, a_1$ for each and the associated rewards, defined as follows:

$$\text{Instance } \textcircled{1}: \begin{cases} a_0 : \frac{1}{2} & \forall t \in \boxed{1}, & \frac{1}{2} & \forall t \in \boxed{2}, & \frac{1}{2} & \forall t \in \boxed{3} \\ a_1 : 0 & \forall t \in \boxed{1}, & 0 & \forall t \in \boxed{2}, & 0 & \forall t \in \boxed{3} \end{cases}$$

$$\text{Instance } \textcircled{2}: \begin{cases} a_0 : \delta & \forall t \in \boxed{1}, & 0 & \forall t \in \boxed{2}, & 0 & \forall t \in \boxed{3} \\ a_1 : 0 & \forall t \in \boxed{1}, & 1 & \forall t \in \boxed{2}, & 1 & \forall t \in \boxed{3} \end{cases}$$

$$\text{Instance } \textcircled{3}: \begin{cases} a_0 : \delta & \forall t \in \boxed{1}, & 0 & \forall t \in \boxed{2}, & 10 & \forall t \in \boxed{3} \\ a_1 : 0 & \forall t \in \boxed{1}, & 1 & \forall t \in \boxed{2}, & 0 & \forall t \in \boxed{3} \end{cases}$$

where phase $\boxed{1}$ is made by the first $T/4$ rounds, phase $\boxed{2}$ is made by the next $T/4$ rounds, phase $\boxed{3}$ is made by the last $T/2$ rounds and $\delta$ is near to $0$.

In phase $\boxed{1}$ all the instances have the same ranking feedback, as the first action gives higher rewards with respect to the second one. To make instance $\textcircled{1}$ receive $R_T \leq CT$, it is necessary:

$$\frac{1}{2}T - \frac{1}{2}\mathbb{E}[n_{a_0}] \leq CT \Rightarrow \mathbb{E}[n_{a_0}] \geq (1 - 2C)T$$

where $n_{a_0}$ is the number of times the first arm has been pulled, and the expected value is taken on the randomization of the algorithm. From previous equation we obtain that in all instances:

$$\mathbb{E}\left[n_{a_0}^{\boxed{1}}\right] \geq (1 - 2C)T - \frac{3}{4}T = (1 - C_1)T/4$$

where $C_1 = 8C$, $n_{a_0}$ is the number of time the first arm has to be pulled in phase $\boxed{1}$ and the inequality is computed considering that $a_0$ is played in all the next phases.
By reverse Markov inequality:

$$\mathbb{P}\left(n_{a_0}^{\boxed{1}} > (1 - \bar{C}_1)T/4\right) \geq \frac{\bar{C}_1 - C_1}{C_1}$$

Setting the probability equal to $9/10$ we obtain:

$$\bar{C}_1 = 10C_1$$

from which follow that with probability $9/10$ we have:

$$n_{a_0}^{\boxed{1}} > (1 - 10C_1)T/4$$

and consequently:

$$n_{a_1}^{\boxed{1}} \leq 10C_1 T/4.$$

We observe that in the second phase, instances ② and ③ have the same feedback. Proceeding as done before, to make instance ② receive $R_T \leq CT$ it is necessary:

$$\frac{3}{4}T - \mathbb{E}\left[n_{a_1}\right] \leq CT \Rightarrow \mathbb{E}[n_{a_1}] \geq \left(\frac{3}{4} - C\right)T$$

From previous equation we obtain that in instances ② and ③:

$$\mathbb{E}\left[n_{a_1}^{\boxed{2}}\right] \geq \left(\frac{3}{4} - C\right)T - T/2 = (1 - C_2)T/4$$

where the inequality is computed considering that $a_1$ is played in the next phases and $C_2 = 4C$. By reverse Markov inequality, we obtain that, with probability $9/10$:

$$n_{a_1}^{\boxed{2}} > (1 - 10C_2)T/4$$

and consequently:

$$n_{a_0}^{\boxed{2}} \leq 10C_2 T/4$$

We neglect the $\delta$ value for now, as it can be chosen to be insignificant with respect to the previous computation.

Now we focus on the third phase, in which instance ② should play:

$$\mathbb{E}\left[n_{a_1}^{\boxed{3}}\right] \geq \left(\frac{3}{4} - C\right)T - T/4 = (1 - C_3)T/2,$$

where $C_3 = 2C$. By reverse Markov inequality, we obtain that, with probability $9/10$:

$$n_{a_1}^{\boxed{3}} > (1 - 10C_3)T/2$$

and consequently:

$$n_{a_0}^{\boxed{3}} \leq 10C_3 T/2$$

Now, we compute the number of rounds needed in the third instance to switch the ranking in the third phase, namely $q$. Notice that, until this switch, the last two instances receive the same feedback. We compute $q$ in the best-case scenario (that is, when small $q$ value is sufficient to allow the switch) that satisfies the constraints previously shown. Precisely, $q$ is computed so that the empirical mean of arm $a_0$ is greater then the arm $a_1$ one, given that $n_{a_0}^{\boxed{1}} > (1 - 10C_1)T/4$ and $n_{a_1}^{\boxed{2}} > (1 - 10C_2)T/4$. Formally:

$$\frac{0(1 - 10C_1)T/4 + 10q}{q + (1 - 10C_1)T/4} \geq \frac{0C_1 10T/4 + (1 - 10C_2)T/4 + 0T/2}{10C_1 T/4 + (1 - 10C_2)T/4 + T/2}$$

We now show that for proper $C$ value we can lower bound the right side with $\frac{1}{4}$. In particular:

$$\frac{0C_1 10T/4 + (1 - 10C_2)T/4 + 0T/2}{10C_1 T/4 + (1 - 10C_2)T/4 + T/2} > \frac{1}{4} \Rightarrow C < 1/200$$

which means that, for $C < \frac{1}{200}$, we can substitute the right side of the equation with $\frac{1}{4}$ to simplify the computation. Moreover, notice that gap between $\frac{1}{4}$ and $\frac{0C_1 10T/4 + (1 - 10C_2)T/4 + 0T/2}{10C_1 T/4 + (1 - 10C_2)T/4 + T/2}$ allowed us to neglect the computations with $\delta$. Then:

$$\frac{0(1 - 10C_1)T/4 + 10q}{q + (1 - 10C_1)T/4} \geq 1/4 \Rightarrow q \geq \frac{4}{39}\left(\frac{1}{4} - 20C\right)T/4$$

To achieve a contradiction, it sufficient to find $C$ so that $q + n_{a_1}^{\boxed{3}} > T/2$; indeed, the previous inequality shows the impossibility to gain enough rewards to make the ranking change and, at the same time, guarantee the minimum rewards to make instance ② no-regret. Given that the ranking switch is a necessary condition to make instance ③ no-regret, the result of impossibility follows for:

$$\frac{4}{39}\left(\frac{1}{4} - 20C\right)T/4 + (1 - 20C)T/2 > T/2 \Rightarrow C < \frac{1}{1640}$$

To conclude the proof, we show that the intersection between the events derived by reverse Markov inequality (namely $E_i$ with $i \in [3]$) holds with constant probability:

$$\mathbb{P}\left(\bigcap_{i\in[3]} E_i\right) = 1 - \mathbb{P}\left(\bigcup_{i\in[3]} E_i^c\right)$$
$$\geq 1 - \sum_{i\in[3]} \mathbb{P}(E_i^c)$$
$$= 1 - \frac{3}{10} = \frac{7}{10}$$

where the inequality holds by Union Bound. Substituting all the previous results in the definition of regret we obtain, with probability $\frac{7}{10} = 1 - \bar{\epsilon}$ and $C < \frac{1}{1640}$, $R_T \geq CT = \Omega(T)$ which concludes the proof. □

# D    Numerical evaluation

This section presents a numerical evaluation of the algorithms proposed in the paper for the *stochastic settings*, namely, DREE and R-LPE. The goal of such a study is to show two crucial results: firstly, the comparison of our algorithms with a well-known bandit baseline, and secondly, the need to develop distinct algorithms tailored for instance-dependent and instance-independent scenarios.

To establish a benchmark for comparison, we consider the EC (Explore-Then-Commit) algorithm, which is one of the most popular algorithms among the explore-then-commit class providing sub-linear regret guarantees. In the following, we evaluate the DREE algorithm with different choices of the $\delta$ parameter in the function $f(t) = \log(t)^{1+\delta}$; precisely, we choose $\delta \in \{1.0, 1.5, 2.0\}$. Furthermore, we consider four stochastic instances whose specific parameters are discussed below. In all these instances, we assume the rewards to be drawn from Gaussian random variables with unit variance, *i.e.*, $\sigma^2 = 1$, and we let the time horizon be equal to $T = 2 \cdot 10^5$. Finally, for each algorithm, we evaluate the cumulative regret averaged over 50 runs.

We structure the presentation of the experimental results into two groups. In the first, the instances have a small $\Delta_{\min}$, while in the second, the instances have a large $\Delta_{\min}$.

**Small values of $\Delta_{\min}$**    We focus on two instances with $\Delta_{\min} < 0.05$. In the first of these two instances, we consider $n = 4$ arms, and a minimum gap of $\Delta_{\min} = 0.03$. In the second instance, we consider $n = 6$ arms, with $\Delta_{\min} = 0.03$. The expected values of the rewards of each arm are reported in Section D.1, while the experimental results in terms of average cumulative regret are reported in Figures 1–2. We observe that in the first instance (see Figure 1) all the DREE algorithms exhibits a linear regret bound, confirming the strong sensitivity of this family of algorithms on the parameter $\Delta_{\min}$ in terms of regret bound. In contrast, the R-LPE algorithm exhibits better performances in terms of regret bound, as its theoretical guarantee are independent on the values of $\Delta_{\min}$. Furthermore, Figure 2 shows that the DREE algorithms (with $\delta \in 1.0, 1.5$) achieve a better regret bound when the number of arms is increased. Indeed, these regret bounds are comparable to the ones achieved by the R-LPE algorithm. The previous result is reasonable as the presence of $\Delta_i$-s in the regret bound lowers the dependence on the number of arms. It is worth noticing that all our algorithms outperform the baseline EC.

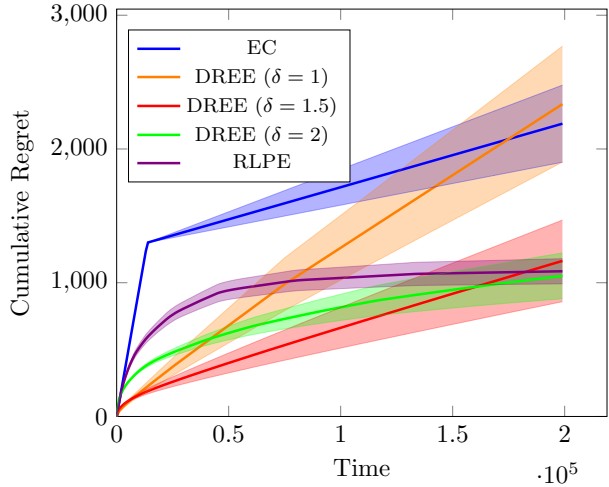

Figure 1: Instance with $\Delta_{\min} = 0.03$ and all the gaps small.

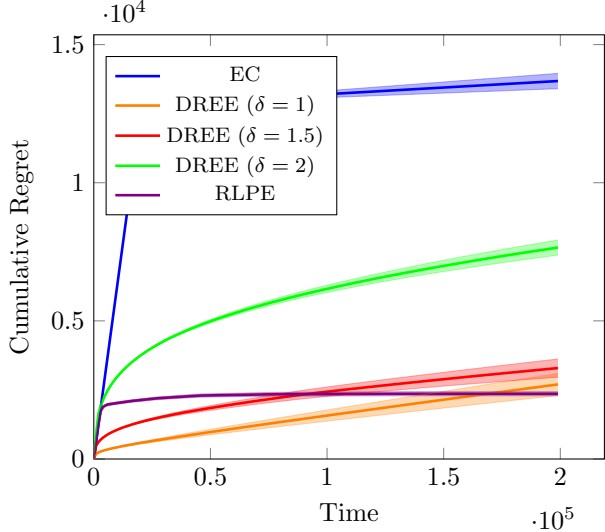

Figure 2: Instance with $\Delta_{\min} = 0.03$ and the other gaps big

**Large values of $\Delta_{\min}$**  We focus on two instances with $\Delta_{\min} \geq 0.25$. In the first instance, we consider $n = 4$ arms with a minimum gap of $\Delta_{\min} = 0.5$ among their expected rewards. In the second instance, we instead consider a larger number of arms, specifically $n = 8$, with a minimum gap equal to $\Delta_{\min} = 0.25$. The expected values of the rewards are reported in Section D.1, while the experimental results in terms of average cumulative regret are provided in Figures 3–4. As it clear from both Figures 3–4 when $\Delta_{\min}$ is sufficiently large, the DREE algorithms (with $\delta \in \{1.0, 1.5\}$) achieves better performances with respect both the EC and R-PLE algorithms in terms of cumulative regret. Furthermore, there is empirical evidence that a small $\delta$ guarantees better performance, which is reasonable according to theory. Indeed, when $\delta$ is small, the function $f(t)$, which drives the exploration, is closer to a logarithm. Also, as shown in Corollary 1, when $\Delta_{\min}$ is large enough, the parameter $\delta$ affects the dimension of $C(f, \Delta_i)$ more weakly, which results in a better regret bound.

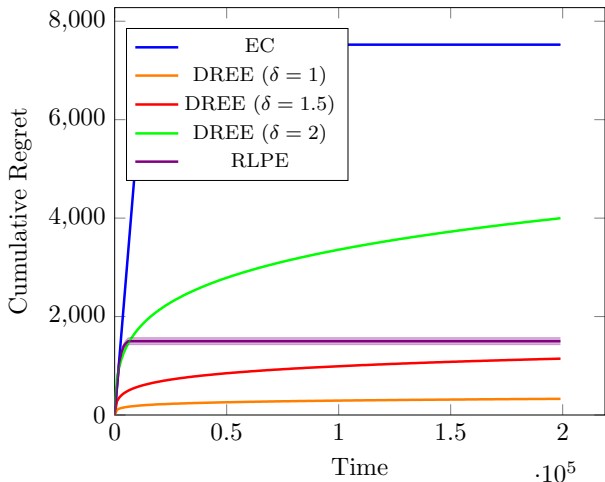

Figure 3: Instance with $\Delta_{\min} = 0.5$.

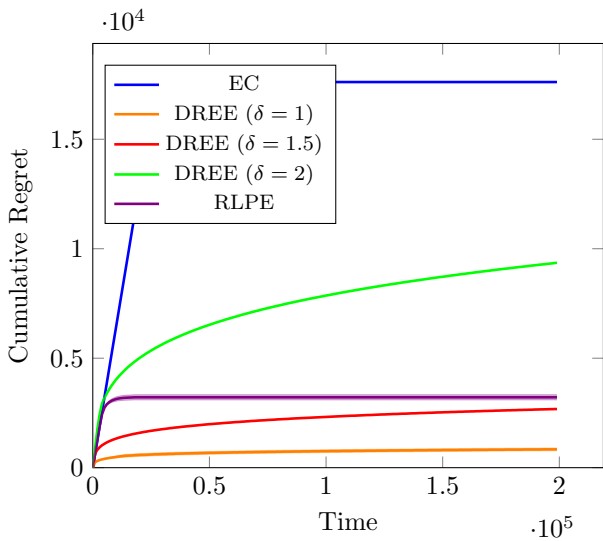

Figure 4: Instance with $\Delta_{\min} = 0.25$.

## D.1 Experiments

For the sake of clarity, we report in the followings additional details on the four instances presented in Figures 1,2,3,4. Notice that, all the plots present the cumulative regret averaged for 50 runs, with 95% confidence interval. Furthermore:

- *Instance of Figure 1*: time horizon $T = 2 \cdot 10^5$, arms $n = 4$, mean reward vector
$$\boldsymbol{\mu} = [0.9, 1.05, 1.12, 1.15],$$
unitary variance for each arm, $\Delta_{\min} = 0.03$;

- *Instance of Figure 2*: time horizon $T = 2 \cdot 10^5$, arms $n = 6$, mean reward vector
$$\boldsymbol{\mu} = [0.03, 0.07, 0.1, 0.08, 0.97, 1],$$
unitary variance for each arm, $\Delta_{\min} = 0.03$;

- *Instance of Figure 3*: time horizon $T = 2 \cdot 10^5$, arms $n = 4$, mean reward vector
$$\boldsymbol{\mu} = [0.05, 0.25, 0.5, 1.0],$$

unitary variance for each arm, $\Delta_{\min} = 0.5$;

- *Instance of Figure 4*: time horizon $T = 2 \cdot 10^5$, arms $n = 8$, mean reward vector

$$\boldsymbol{\mu} = [0.05, 0.05, 0.1, 0.15, 0.25, 0.5, 0.75, 1.0],$$

unitary variance for each arm, $\Delta_{\min} = 0.25$;

## D.2  Detailed explanation of the experiments

In this section, we report all the details of the experiments performed in the paper. These are important to ensure the truthfullness of the results and the claims based on empirical validation.

**Training details**   In the main paper we have presented four experiments, each corresponding to a different environment. Each experiment is performed for fifty random seeds, ad the computation is split in 10 parallel processes by the library `joblib`. The overall computational time for one experiment is around $337.92$ seconds, that is roughly five minutes and one half.

**Compute**   As stated, the numerical simulations resulted to be very fast. For this reason, it was not necessary to run them on a server, and we used a personal computer with the following specifications:

- CPU: `11th Gen Intel(R) Core(TM) i7-1165G7 2.80 GHz`
- RAM: `16,0 GB`
- Operating system: `Windows 11`
- System type: `64 bit`

**Reproducibility**   Due to the stochastic nature of the bandit problem, all the simulations have been repeated several times. We have performed all the experiments with $50$ different random seeds, corresponding precisely to the first $50$ natural numbers. The seed influences the generation of the reward by the environment, while all algorithms proposed, being deterministic, are independent on the seed.

