# OpenReview forum: "Bandits with Ranking Feedback"
_NeurIPS.cc/2024/Conference — NeurIPS 2024 poster_

### Official Review · Reviewer_cfwJ · 2024-06-12

**Soundness:** 3
**Presentation:** 3
**Contribution:** 3
**Rating:** 7
**Confidence:** 5

**Summary:**

The paper studies the multi-armed bandit problem when the learner only observes the ranking of the average cumulative reward of all the arms. This is a strictly worse information environment than the standard setting. The paper proposes algorithms that still attains the instance independent and dependent regret rates that match the stochastic multi-armed bandit. It also shows that no algorithms can do better in both the instance-dependent and -independent settings simultaneously.

**Strengths:**

- The setting studied in the paper is novel. I have not read similar setups in the literature before. Although the practical motivation can be strengthened, I think it is an interesting problem to study. One potential application could be some kind of tounament in which only the ranking is observed.
- Although the information is a lot less than the standard MAB, the authors show that the optimal regret in both cases (instance-dependent and -independent) can still be achieved. This is surprising. The tools used are standard probabily theory, which is a plus to me.
- The trade-off between instance-independent and -dependent regret is interesting and surprising. This is not the case in standard MAB and I haven't seen similar results before.

**Weaknesses:**

- It is probably out of the scope of this paper. But the current dependence on $n$, the number of arms, is quite far from optimal ($n^4$) in the instance dependent case. I wonder if the authors have thought about better designs to obtain linear or sublinear dependence. Some discussion would be helpful.

**Questions:**

- In the proof of Theorem 4, the policy doesn't seem to depend on $T$. For example, in line 534, the probability of $\pi(pull 2|E_\tau)$ is a function of $\tau$ only. My understanding that the algorithm is allowed to depend on $T$. Does the proof work for this case?
- I think the title misleads the readers. Ranking feedback sounds like receiving the order of the rewards in the current round. It is up to the authors, but a more informative title would help the paper.

---

> ### Author Rebuttal · Authors · 2024-08-06
>
> Q: _It is probably out of the scope of this paper. But the current dependence on $n$, the number of arms, is quite far from optimal in the instance dependent case. I wonder if the authors have thought about better designs to obtain linear or sublinear dependence. Some discussion would be helpful._
>
> The fact that our instance-independent bound in Theorem 8 scales with $ n^4 $ is one of the most interesting open points of this paper. The reason for this unusual order may be found in the fact that the ranking setting prevents us from relying on standard concentration properties. Therefore, no "high-probability" estimates can be performed, and each time in the analysis we perform a "union bound" over the space of arms, this results in an additional $ n$ factor in the regret bound. In contrast, when using a high-probability estimate, the same union bound would result in an additional $\log(n)$ term, which is negligible. Furthermore, given the ranking feedback setting, it appears natural that at least two union bounds over the space of arms must be performed in order to compare any possible pair of arms. Therefore, we **believe** that a super-linear dependence on $ n $ in the instance-independent regret bound cannot be avoided, but we still **conjecture** that such dependence may be moved to lower order terms.
>
> _Q: In the proof of Theorem 4, the policy doesn't seem to depend on $T$. For example, in line 534, the probability of $\pi(pull|E)$ is a function of $\tau$ only. My understanding that the algorithm is allowed to depend on $T$. Does the proof work for this case?_
>
> The point made by the Reviewer is correct, and very subtle. In fact, the current lower bound only applies to any-time policies, independent of the time horizon. Fortunately, modifying the proof to generalize the result to the case where the policy can depend on $T$ is not difficult.
> We start the proof by contradiction, assuming that both
>     $$R_T(\pi_T)\le C(\Delta)T^\alpha \qquad R_T(\pi_T)\le T^\beta$$
>
> hold for any $T$, given a sequence of policies $\pi_T$. The "hard instances" are the same, both with two arms and small $\Delta$, and also the event $E_t$ is defined as in the paper. From the assumption, it follows that
>
> $$C(\Delta= 1)T^\alpha = C( 1)T^\alpha \ge \mathbb E[Z_2(T)|E_T],$$
>
> otherwise the sequence of policies would suffer regret more than $C( 1)T^\alpha$ in case arm one gives always $1$ and the other always $0$. From this equation, it is easy to complete the proof as usual.
>
> In the original proof, the only step requiring the policy to be any-time was the limit
> $$\forall \eta>0,\ \limsup_{t\to \infty}\frac{\sum_{\tau=1}^t \pi(\text{pull 2}|E_\tau)}{t^{\alpha+\eta}}=0,$$
>
> which is not necessary and can be easily avoided working by contradiction. We will put this improved proof in the final version of the paper, we remain at the Reviewer disposal in case they need further clarification on how to change this proof.
>
> _Q: I think the title misleads the readers. Ranking feedback sounds like receiving the order of the rewards in the current round. It is up to the authors, but a more informative title would help the paper._
>
> We thank the Reviewer for the interesting observation. Nevertheless, we believe the feedback structure the Reviewer is describing is somewhat referred to as "Dueling bandits", or their generalization. Thus, we are open to any suggestion.

---

> > ### Comment · Reviewer_cfwJ · 2024-08-09
> >
> > Thanks for the detailed response. I don't have further questions.

---

### Official Review · Reviewer_KUAx · 2024-07-08

**Soundness:** 4
**Presentation:** 4
**Contribution:** 4
**Rating:** 7
**Confidence:** 3

**Summary:**

The paper studies the setting where, every time an arm is pulled, a principal gets to observe a reward, but the player only gets to observe the order that emerges from the accumulated rewards so far. The authors study both the adversarial and stochastic, and both the instance-dependent and instance-independent regrets. They provide tight results for all cases, with the only exception being the stochastic instance-independent case, where they provide tight bounds only for the Gaussian model.

**Strengths:**

- The paper is well-written. Intuitive explanations of the results are provided along with a nice description of the algorithms.

- The model is super interesting in my opinion, at least from a mathematical point of view, and quite novel.
-  The authors provide a very complete picture. They study both the adversarial and non-adversarial settings, as well as instance-dependent and instance-independent regrets.

- Overall, I really enjoyed reading the paper

**Weaknesses:**

- I miss some more realistic motivation for the model and some concrete applications. As I said, from a theoretical point of view, the model is very interesting, but when I tried myself, I could not come up with a clear application.

- For the general stochastic and instance-independent case, the only guarantee that is provided is that of the EC algorithm which can be easily applied in this setting

- R-LPE algorithm needs to know T

**Questions:**

No questions

**Limitations:**

No limitations

---

> ### Author Rebuttal · Authors · 2024-08-06
>
> Q: _I miss some more realistic motivation for the model and some concrete applications. As I said, from a theoretical point of view, the model is very interesting, but when I tried myself, I could not come up with a clear application._
>
> We thank the Reviewer for the question. In the following, we provide a real-world example of a possible application of our setting. In pay-per-click online advertising (the total spent is of the order of several billion USD per year), large platforms optimize advertisers' campaigns. Specifically, these platforms observe the number of clicks of each single campaign, but to allocate the budget most effectively (using a knapsack-style approach), they need to know the revenue of the individual campaigns. Obviously, the platforms cannot observe the revenue, which is private information of the advertiser. On the other hand, advertisers do not want to communicate this private information to the platforms, and, for this reason, the platforms limit themselves to maximizing the number of clicks. However, this kind of optimization leads to very approximate solutions compared to considering the revenue as well. The use of bandits with ranking feedback  in this context would circumvent this problem. In particular, advertisers would be asked for feedback on the ranking of advertising campaigns, avoiding the need to ask for revenue information.
>
> Q: _For the general stochastic and instance-independent case, the only guarantee that is provided is that of the EC algorithm which can be easily applied in this setting_
>
> We thank the Reviewer for the comment but we believe there is a potential misunderstanding on our results. Indeed, the **R-LPE algorithm is specifically tailored for the instance independent case** while it achieves a subotptimal instance dependent regret bound. Finally, please notice that R-LPE instance independent regret guarantees are far better than those of the standard Explore and Commit algorithm which are of the order $\mathcal{O}(T^{2/3})$.
>
> Q: _R-LPE algorithm needs to know $T$_
>
> The reviewer is correct. In our settings, due to the specific nature of the feedback, we cannot employ the well-known "doubling trick" to relax the knowledge of the time horizon $T$. We leave it as an interesting open problem to determine whether the requirements on the knowledge of time horizon $T$ can be relaxed.  For what concerns the empirical validation, in the attached PDF we added an experiment measuring the impact of using a misspecified value for $T$.

---

> > ### Comment · Reviewer_KUAx · 2024-08-09
> > **No further questions**
> >
> > I would like to thank the authors for their answer and the clarifications. I do not have any further questions at this point. While I still do not find the motivating example very convincing, I will keep the score as it is.

---

### Official Review · Reviewer_iUMy · 2024-07-15

**Soundness:** 3
**Presentation:** 2
**Contribution:** 3
**Rating:** 6
**Confidence:** 3

**Summary:**

The paper introduces a variant of the multi-armed bandit problem called "bandits with ranking feedback," where the feedback ranks the arms based on historical data without showing precise numerical differences. This approach is particularly useful in scenarios where exact measurement of values is impractical, such as with human preferences or confidential data. The main contributions of the study include developing no-regret algorithms that operate under both stochastic and adversarial conditions for this model. The findings indicate that achieving logarithmic regret is impossible with ranking feedback in the stochastic setting, and no algorithm can achieve sublinear regret in the adversarial setting. The paper proposes two algorithms: DREE, which achieves superlogarithmic regret in stochastic instances, and R-LPE, which manages a regret of O(\sqrt{T})in stochastic instance-independent scenarios. These innovations significantly enhance the understanding and implementation of bandit algorithms in complex feedback environments.

**Strengths:**

Quality: The theoretical contributions are robust, including the proof that no algorithm can achieve logarithmic regret in the stochastic setting with ranking feedback, and no sublinear regret is achievable in the adversarial setting
Clarity: The paper is well-structured, with clear delineation of problem settings, algorithmic approaches, and theoretical analyses.

**Weaknesses:**

1) Originality: Is the concept of "bandits with ranking feedback" truly novel?
2) Experimental Validation: The paper would benefit from more experimental validation. While the theoretical aspects are well-developed, further empirical testing in diverse conditions could strengthen the validation of the algorithms. Tests with non-Gaussian noise and in real-world settings would be particularly insightful.
3) Algorithm Complexity: The discussion on the computational complexity and practical scalability of the introduced algorithms is limited. More detailed analysis in this area could provide better insights into their applicability in real-world scenarios.
4) Adversarial Setting Analysis: It is mentioned that no algorithm can achieve sublinear regret in adversarial settings, but more detailed explanations or suggestions for alternative approaches to handle such conditions would be beneficial.
5) Impact of Dependence on Parameters: The algorithms appear to heavily rely on the correct setting of specific parameters, such as the time horizon T. Discussing the sensitivity of the algorithms to these parameters and providing strategies for effective parameter tuning would aid their practical application.
6) Extension to Other Models: Could the principles of ranking feedback be applied to other types of bandit problems, such as contextual bandits or those with structured action spaces? This extension could broaden the applicability of the research findings.

**Questions:**

1) Originality: Is the concept of "bandits with ranking feedback" truly novel?
2) Experimental Validation: The paper would benefit from more experimental validation. While the theoretical aspects are well-developed, further empirical testing in diverse conditions could strengthen the validation of the algorithms. Tests with non-Gaussian noise and in real-world settings would be particularly insightful.
3) Algorithm Complexity: The discussion on the computational complexity and practical scalability of the introduced algorithms is limited. More detailed analysis in this area could provide better insights into their applicability in real-world scenarios.
4) Adversarial Setting Analysis: It is mentioned that no algorithm can achieve sublinear regret in adversarial settings, but more detailed explanations or suggestions for alternative approaches to handle such conditions would be beneficial.
5) Impact of Dependence on Parameters: The algorithms appear to heavily rely on the correct setting of specific parameters, such as the time horizon T. Discussing the sensitivity of the algorithms to these parameters and providing strategies for effective parameter tuning would aid their practical application.
6) Extension to Other Models: Could the principles of ranking feedback be applied to other types of bandit problems, such as contextual bandits or those with structured action spaces? This extension could broaden the applicability of the research findings.

**Limitations:**

Experimental Scope: The authors could extend the experimental validation to include real-world datasets or scenarios to better demonstrate the practicality of the algorithms under different environmental conditions.

---

> ### Author Rebuttal · Authors · 2024-08-06
>
> Q: _Originality: Is the concept of "bandits with ranking feedback" truly novel?_
>
> To the best of our knowledge, the "bandits with ranking feedback" model introduced in our paper represents a new bandit setting. Although our setting shares similarities with dueling bandits, the two settings are substantially different, as we discuss in the related works section in our paper.
>
> Q: _Experimental Validation._
>
> Following the suggestion of the reviewer, we enriched our experimental evaluation, and the new results we obtained can be found in the attached PDF. In particular, we investigated the following:
>
> - **Non Gaussian noise.** In the attached PDF, we ran the same experiments of the paper with uniformly distributed noise. We observe that this leads to similar results. However, the variance of the regret curves is slightly increased, but the mean remains nearly the same.
>
> -  **Measuring the computational effort required by the algorithms.** While the running time of DREE is clearly linear in $T$, the running time of R-LPE is less straightforward to compute. However, in the additional experiments, we empirically observed that the running time of the R-LPE algorithm is also linear in $T$.
>
> - **Robustness of R-LPE to misspecification of $T$.**  Even if the algorithm strongly relies on knowledge of the time horizon, we have shown that providing the algorithm with a time horizon larger than the actual one is not particularly harmful. On the other hand, as is usual in bandit algorithms, setting the time horizon $T'$ smaller than the actual time horizon $T$ results in linear growth of the regret after $T'$ rounds.
>
> Q: _Algorithm Complexity._
>
> We thank the Reviewer for underlining this important aspect, which we did not mention in the paper due to space constraints. Both our algorithms are very computationally efficient. In the experiment, we have empirically demonstrated this fact, and we will show it theoretically in this rebuttal. Specifically, the **DREE** algorithm requires either pulling the first-ranked arm (in most rounds) or an arm in a lower position of the ranking according to a deterministic schedule. Therefore, it does not require computing confidence regions, resulting in a running time of just $\mathcal{O}(nT)$. The second algorithm, **R-LPE**, is more complex compared to DREE, as it requires updating a set of active arms $S$ at certain rounds. Fortunately, this set is only updated during the time steps that belong to the loggrid, and thus it is updated a logarithmic number of times. Therefore, even though there is a summation over $t$ in the definition of the set $S$, the computational complexity of R-LPE is not quadratic in $T$, but rather of order $\mathcal{O}(T+nT\log(T))$. The first term in the latter expression corresponds to the "usual" rounds, where we simply follow a round-robin strategy. The second term corresponds to the product of $\log(T)$ (the number of rounds in the loggrid), $T/n$ (the number of "fair" rounds), and $n^2$ (the number of possible comparisons between pairs of arms).
>
> Q: _Adversarial Setting Analysis._
>
> We thank the Reviewer for the question.
>     **On the results explanation.** To derive our adversarial lower bound we considered three instances. The instances are similar in terms of rewards for the first two phases but differ significantly in the third one. Thus, we show that, if the learner receives the same ranking when playing in two instances with different best arms in hindsight, it is impossible to achieve low regret in both scenarios.
>
> **On possible alternative approaches.** We agree with the reviewer that, to handle such cases, different metrics could be taken into consideration. For instance, it would be possible to study algorithms achieving sufficiently "good" competitive ratios, namely, those that are no-regret with respect to a large fraction of the optimum. We leave the aforementioned research directions as interesting future work.
>
> Q: _Impact of Dependence on Parameters._
>
> The R-LPE algorithm requires knowledge of the time horizon $T$, while the DREE algorithm is an anytime algorithm. In the attached PDF we added an experiment measuring the empirical impact of using a misspecified value of the time horizon $T$. We would like to emphasize that the R-LPE algorithm has no additional hyper-parameters, since the quantities that characterize it, such as the loggrid and the parameter $\alpha$, can be computed as a function of $T$. Additionally, while the definition of $\alpha$ at Line 8 could be tuned experimentally, doing so may prevent us from achieving the desired regret guarantees of $\mathcal{O}(\sqrt{T})$, which is the primary goal of our paper.
>
> Q: _Extension to Other Models_
>
> Extending our results to settings where the action space is no longer discrete is highly non-trivial, as it would first require rethinking the concept of ranking feedback. However, if we consider linear (or possibly contextual) bandits with finitely many arms, it would be interesting to see if such a linear structure allows for a better dependence on the number of arms (possibly sublinear as in [1]) in the instance indepenednt regret bound. Nonetheless, it should be noted that, from a technical perspective, combining linear settings with ranking ones would require merging the argument based on optimal design for the least squares estimator [2] or self-normalized processes [3] with our bound, which is instead based on Levy's arcsin law.
>
> [1] Tor Lattimore, Csaba Szepesvari, Gellert Weisz. Learning with Good Feature Representations in Bandits and in RL with a Generative Model.
>
> [2] Kiefer and J. Wolfowitz. The equivalence of two extremum problems.
>
> [3] Yasin Abbasi-yadkori, Dávid Pál, Csaba Szepesvári, Improved Algorithms for Linear Stochastic Bandits

---

### Official Review · Reviewer_hpCM · 2024-07-18

**Soundness:** 3
**Presentation:** 2
**Contribution:** 2
**Rating:** 5
**Confidence:** 2

**Summary:**

This paper considers a new multi-armed bandit problem where the feedbacks are ranking of the arms. In this problem, the environment gives the feedback on the ranking of the arms based on the previous pulls. The authors first consider the stochastic setting and give the lower bound of the regret for the instance-dependent case. Then, they propose a design of explore-then-commit, called DREE, which is proved to achieve a sublunar regret for instance-dependent case. After discussing the regret tradeoff between instance-dependent and  instance-independent cases, the authors design a phase elimination algorithm (R-LPE) that has a sublunar regret. Then they move beyond the stochastic setting and prove that sub-linear regret cannot be achieved for the adversarial setting.

**Strengths:**

The paper has the following strengths.
+ The paper considers a novel setting of bandits with ranking feedback. The ranking feedback depends on the history of reward,  so it is different from the dueling bandits.
+ For the stochastic setting, the paper derives the lower regret bound to show the difficulty of the instance-dependent case.
+ The paper proposes provable algorithms to solve bandits with ranking feedback for both  instance-dependent and instance-independent cases. Adequate analysis is provided to support their points.

**Weaknesses:**

I am concerned about the following aspects.
First, in the problem setting, the ranking feedback is assumed to be perfect in terms of the ranking of the averaged history rewards. However, human probably not give the perfect ranking since history reward is not easily observed. Thus, it would be better if the authors can discuss the designs with imperfect ranking feedback.
Second, the analysis in the adversarial setting does not give any insight. It is obvious that the regret is linear for adversarial bandits with ranking feedback. However, can the authors discuss whether their algorithms are robust enough for adversarial setting? (e.g. provide an analysis of the competitive ratio.)

**Questions:**

Can the authors give more concrete examples to motivate the setting of bandits with ranking feedbacks?
Can the authors provide a high-level explanation on why there exists a tradeoff between instance-dependent and instance-independent cases?

**Limitations:**

The authors do not include a discussion of the limitations. However, a discussion on the limitations of the setting will be helpful.

---

> ### Author Rebuttal · Authors · 2024-08-06
>
> Q: _"First, in the problem setting, the ranking feedback is assumed to be perfect in terms of the ranking of the averaged history rewards. (...)"_
>
> We completely agree with the Reviewer that the introduction of imperfection/uncertainty/tolerance is of paramount importance as it would allow to capture better the actual human behavior, and this refinement is exactly the next step of our agenda. On the other side, no imperfection can be introduced before a complete study of the perfect model, and, as we show in our paper, the study of the exact model is not straightforward. Furthermore, we observe that introducing a perturbation in the ranking observed by the learner may hinder the possibility of designing a no-regret algorithm. Thus, exploring scenarios where the learner receives corrupted ranking feedback and still manages to design a no-regret algorithm is both a fascinating and challenging research direction. Nonetheless, it requires a different approach from the one presented in our work and is something we plan to investigate in the future.
>
> Q: _Second, the analysis in the adversarial setting does not give any insight. (...)_
>
> We believe that the impossibility result for the adversarial setting represents a first yet fundamental step toward a full understanding of the problem when the rewards are adversarially chosen. We are disappointed that the Reviewer found our result trivial, as it is uncommon in the online learning literature to achieve a positive result in the stochastic setting that cannot be extended to the adversarial setting. This feature is peculiar to the bandits with ranking feedback model, as even related settings, such as dueling bandits, do not exhibit this kind of impossibility result. Finally, we agree with the Reviewer that studying the competitive ratio of algorithms developed for adversarial settings is an interesting research direction, and we plan to pursue it in the future.
>
> Q: _Can the authors give more concrete examples to motivate the setting of bandits with ranking feedbacks? Can the authors provide a high-level explanation on why there exists a tradeoff between instance-dependent and instance-independent cases?_
>
> We thank the Reviewer for the question. In the following, we provide a real-world example of a possible application of our setting. In pay-per-click online advertising (the total spent is of the order of several billion USD per year), large platforms optimize advertisers' campaigns. Specifically, these platforms observe the number of clicks of each single campaign, but to allocate the budget most effectively (using a knapsack-style approach), they need to know the revenue of the individual campaigns. Obviously, the platforms cannot observe the revenue, which is private information of the advertiser. On the other hand, advertisers do not want to communicate this private information to the platforms, and, for this reason, the platforms limit themselves to maximizing the number of clicks. However, this kind of optimization leads to very approximate solutions compared to considering the revenue as well. The use of bandits with ranking feedback  in this context would circumvent this problem. In particular, advertisers would be asked for feedback on the ranking of advertising campaigns, avoiding the need to ask for revenue information.
>
> Q: _Can the authors provide a high-level explanation on why there exists a tradeoff between instance-dependent and instance-independent cases?_
>
> Usually, in multi-armed bandits, an instance-independent regret bound can be derived from an instance-dependent one. Indeed, if the instance-dependent regret bound depends on the suboptimality gaps as $ R_T \propto {\log(T)}/{\Delta} $, it is possible to take the worst possible value of $ \Delta $ and achieve the desired instance-independent regret bound. Unfortunately, in our case, the instance-dependent regret bound cannot be written in this form as a consequence of the feedback characterizing our setting. Indeed, we can only observe switches in the ranking, which do not reflect the actual differences in the expected rewards of the arms. Thus, the only way to “explore” in our setting is by pulling arms that could be **highly** sub-optimal. For this reason, achieving an asymptotic regret bound that is close to logarithmic requires exploiting the arm being ranked first several times, thus suffering a particularly unpleasant dependence on the suboptimality gaps $ \Delta_i $ (see, for example, Corollary 3).  Finally, we remark that the formalization of this reasoning is presented in Theorem 4 (Instance Dependent/Independent Trade-off).

---

> > ### Comment · Reviewer_hpCM · 2024-08-12
> >
> > I have read the response. Thank you authors for answering my questions.

---

### Author Rebuttal · Authors · 2024-08-06

Dear Reviewers,

in the attached PDF, we provide additional experiments.

The authors.

---

### Decision · Program_Chairs · 2024-09-25

**Decision:**

Accept (poster)

**Comment:**

The reviewers overall favor the paper's contributions, among which are the introduction of a bandit model with a new form of rank-ordered feedback, and a comprehensive study of attainable performance in this model. I must add, however, that there were also concerns raised almost unanimously about the practical relevance of this setting in modeling applications. The author(s) responded to this by providing a setting involving pay per click online advertising optimization where an agent must optimize for ad displays without knowing (potentially sensitive and private) revenue information per ad. This appears to have been satisfactorily accepted as a motivating example by the reviewers. Noting that the study of this new model can hold potential and inform decision making algorithms for future purposes, I recommend acceptance of the paper.